# Defect Repair Cost and Home Warranty Deposit, Korea

**Junmo Park** [ID] **and Deokseok Seo** *

School of Architecture, Halla University, Wonju-si 26404, Gangwon, Korea; adviser.cm@gmail.com
* Correspondence: seodk@halla.com; Tel.: +82-10-2289-9946

**Abstract:** Defects in the home cause inconvenience to homeowners and loss to the developer or builder due to cost and damage to reputation. A home warranty deposit exists to protect consumers from home defects. However, it is unclear how much it costs to repair a defect and whether the standard for calculating the warranty deposit is reasonable. This study analyzed the litigation details of 290 home complexes in Korea to investigate defect repair costs and factors affecting them. According to the analysis results, the defect repair cost was 0.538% of the construction cost on average. This is far below 3% of the construction cost, the prevailing standard in the relevant acts. Additionally, there was no case where the defect repair cost ratio exceeded 3%. The actual repair cost for defects was considerably lower than the warranty deposit standard stipulated by the Act. Therefore, this study proposes a method to adjust the warranty deposit collectively and a means to apply it differentially according to the size of the construction cost. This proposal benefits everyone because it protects homeowners and keeps the costs affordable. In addition, it is expected that the warranty deposit can be used as a standard for home construction and post-quality improvement. After analyzing the factors affecting the repair cost, it was found that there was a difference in the repair cost according to the home construction implementor and the construction purchase order method. The warranty deposit system prepares for the possibility that a project owner or builder, who is the project implementor, becomes insolvent. It was found that there was no difference in the repair cost for defects depending on whether the project owner was insolvent. Nonetheless, the possibility of insolvency in the case of a developer was far higher than in the case of a builder.

**Keywords:** defect repair cost; home warranty deposit; quality management



## 1. Introduction

Disputes over construction cost, construction period, and quality in the construction sector, including homes, continue [1]. Today, economic and industrial development is advanced, and disputes are intensifying regarding quality issues [2]. Disputes over the quality of homes might have been caused by various factors such as home policies, economic systems, and construction production methods. Since the industrial revolution, cities have been expanding, the population has increased, and the demand for homes has also increased [3]. A stable and continuous supply of homes is necessary to maintain people's lives engaged in economic activities in the city. However, the supply of homes is always short due to fast urbanization and industrialization worldwide. From a global perspective, even the Organization for Economic Cooperation and Development (OECD) average has a lower number of homes per capita than the European Union, and Korea ranks among the lowest [4,5]. Therefore, many countries are sticking to policies to increase home supply. However, the supply and demand of professional manpower, equipment, and materials to support the home supply have always been insufficient [6,7], and some are of the opinion that the number of defects in a home increased as a result [8].

In addition, the production system of the construction industry, including homes, was a producer-centered management method targeting the construction period and construction costs. Specifically, a prevailing trend emphasizes the construction period and

construction cost but places little emphasis on quality. However, the trend has changed significantly in recent years as it has shifted towards consumers who value quality [9,10]. As conflicts over the quality of homes escalate, disputes intensify. Since disputes over home quality are not disputes between project implementors, including developers, designers, builders, and inspectors, but somewhat between these participants and the homeowners, i.e., consumers, the statuses are different. Developers mostly want to resolve conflicts during the construction stage and are not interested in problems that arise after the home is completed. On the other hand, homeowners are inevitably interested in the problems that arise after moving in because the house itself is a tremendous asset and serves as their shelter.

Until the home is handed over to the homeowner, most of the failures or defects discovered in advance by the builder are repaired. However, in many cases, various defects occur or are discovered even after the home is handed over to the consumer. Such a defect is highly unpleasant for the homeowner and is inconvenient because it needs to be repaired. The developer must also bear the time and cost of the maintenance work, so the loss is aggravated. In addition, if defects are found at an early stage after the home's handover, this can be regarded as negligence in the production stage, but it is difficult to conclude that all problems that appear after several years are solely the project owner's responsibility. Meanwhile, the quality problem of new homes has become a social issue in many countries.

In the UK, 57% of new home buyers are dissatisfied with quality [11], and recently built homes are perceived as having many defects, so they are being avoided [12]. In the United States, home warranty companies went bankrupt due to poor home quality [13]. In Australia, home warranty companies went bankrupt or were expelled from the market [14]. In Japan, designers' embezzlement through design manipulation has led to legal battles where the designer was punished and some of the related real estate and builders went bankrupt, while the homeowners did not receive any compensation for their repairs [15]. In Korea, 37,116 home defect disputes were filed between 2010 and 2021 by the Defect Review and Dispute Mediation Committee under the Ministry of Land, Infrastructure, and Transport, and the number of disputes is increasing every year [16]. In addition, in January 2022 in Korea, the construction method of reinforced concrete was arbitrarily changed at the high-rise apartment construction site, and the slab and exterior wall collapsed consecutively during construction, causing a major accident resulting in six casualties. Therefore, concern is growing among homeowners about the quality of homes built by that builder [17].

As the home quality problem becomes more serious, various systems are in place to compensate for the quality problem. Among them, the home warranty deposit (HWD) is used to prepare for defects or failure caused by the negligence of the developer and builder, who are the main project parties [18]. More precisely, the warranty deposit is not something anyone can withdraw and use to repair defects in a home. From the point of view of contract law, since the developer has contracted to supply the home to the homeowner, the developer is responsible for any quality problems or defects in the homes. Of course, it is common for home construction to be done by a builder, a general contractor, or a specialty contractor who has a contract with the developer rather than directly by the developer. Therefore, it is often the builder's responsibility to repair the home's defects practically. In principle, the developer is responsible for his repairs, and the home warranty deposit is for the case of the developer who may become bankrupt, insolvent, or is in need of court management [19].

If so, how often does a developer become insolvent? As in any other industry, the home construction industry has its ups and downs. In particular, cases such as bad money circulation, lack of sales, and disputes with subcontractors happen frequently. The home developer often gets involved in insolvency, such as court mismanagement, dishonor, and bankruptcy. Bankruptcies of project owners such as developers and builders have been standard worldwide, and even in the recent 2010s, they occurred continuously in each country. In Japan, 10 out of 249 general construction companies were out of the market between 2009 and 2010 [20]. In Australia, mid-to-large builders such as Kell and Rigby; St.

Hilliers Construction Pty. Ltd.; Hastie Group Limited; and Southern Cross Construction went bankrupt between 2011 and 2013 [21]. In the United States, 100-year-old Alan Reeves filed for bankruptcy [22]. In the UK, it was also reported that the debt of Carillion, the second-largest company, was at risk due to a snowball of debt [23].

In 2021, amid the corona pandemic, the bankruptcy of real estate companies in Japan increased by 12.5% due to the economic crisis, and 22 construction companies went bankrupt [24]. In the case of China, there were even reports that the China Evergrande Group, a real estate conglomerate, was on the verge of bankruptcy [25]. In Korea, according to the statistical data of the Korea Construction Association, an organization of general construction companies, an average of 67.85 companies went bankrupt over a span of 20 years, from 2001 to 2020 [26]. Of course, there were many bankruptcies in the 2000s, and in the 2010s, they decreased to 1/10 of the level of the 2000s. However, as in the other countries reviewed above, the situation is highly likely to change remarkably when the impact of the prolonged coronavirus pandemic begins to be reflected. It is not clear quantitatively to what extent the home developers become insolvent. Moreover, the incompetence of the home developers is an ever-present risk. A home warranty deposit for repairs must be prepared to minimize the impact of the damage on homeowners.

Then, to what extent should the warranty deposit for repairing defects in new homes be prepared if the developer becomes insolvent? Since the purpose of the warranty deposit is to prepare for defect repair, it is necessary to consider the cost of repairing the defect. According to preceding studies, the cost of repairing defects is, on average, at least 0.18% to at most 5.4% of the construction cost. One may examine the ratio of the repair cost used for rework against construction cost during the construction of a facility or for repairing defects after completion as a basis. According to the study by Josephson et al., an average of 4.4% of the construction cost was proposed as a defect repair cost [27]. The study by Mill et al. proposed an average of 4% for repairing the defect [28]. In the study by Hwang et al., repair costs of an average of 5.4% of the developer's project and 2.2% of the contract project were analyzed [29]. The following details were the reported repair or rework cost: in Choi's study, an average of 1.1% [30]; in Forcada et al.'s study, an average of 2.75% [31]; in Love et al.'s study, an average of 0.18% [32]; and in Liu et al.'s study, an average of 4.95% [33]. These preceding studies show in-depth consideration and outstanding results, although the research objects' facilities were diverse, and data collection was not accessible. However, it is unclear how the repair cost was calculated, and there seems to be a limitation that the sample size (i.e., the number of homes included in the case study) was relatively small.

Therefore, data on repair costs concluded in lawsuits that raised quality issues for newly built homes in Korea were collected in this study. The lawsuit was limited to the case where a lawsuit was filed after the home was completed and handed over to the owner, and the repair cost calculated in the lawsuit was based on the consistent method set by the court. The ratio of the defect repair cost of each home to the construction cost was calculated, and this was compared with the results of related laws and previous research. Through this, a reasonable level of defect repair deposit standards was sought. In addition, factors related to the execution method of the home construction project and to the developer's inability, which is the fundamental reason for the establishment of the warranty deposit system, were analyzed. After that, theoretical reviews were made, followed by detailed explanations of the research methods. Then, case analysis and discussion proceeded, and the conclusion was drawn.

## 2. Literature Study

### 2.1. Defect Warranty Deposit

#### 2.1.1. Outline

Home warranty deposits can be generally divided into cases where the subject home is a newly built one and a case presented by the seller for the buyer after the existing warranty of the home expires. The home warranty of a newly built home is a system

where the developer builds the house and guarantees against any problems (defects, failure, etc.) that occur for a certain period. It is divided into structural factors, architectural frameworks that affect the safety of newly built homes, and other facilities and finishes. Although the home warranty deposit differs from country to country, it is necessary to secure a warranty deposit to guarantee the surety in an authorized financial institution or a company specializing in guaranteeing or purchasing insurance under the condition that the deposit can be secured and issued an insurance policy. On the other hand, when the homeowner or real estate company sells the home, the structural part and various facilities are guaranteed for a specified period.

In most cases, the existing home warranty is operated by insurance. However, even if such a warranty system exists, it cannot solve all the problems in the home, and the limit of liability is clearly stated. A home warranty makes it compulsory for the developer to perform repairs within the warranty period, whether on a newly built home or an existing home. In addition, it is possible only in special cases to be paid a repair cost instead of shouldering repairs or compensating for direct damage caused by a problem.

### 2.1.2. Regulations

In most major countries, it is stipulated that the developer who builds a house should be held responsible for the warranty for home defects. In the United Kingdom and the United States, where the private sector led the way to improve home quality and activate the warranty system, the law does not specify a home warranty deposit for repairs against defects. Instead, they require the home's seller and buyer a warranty contract through individual insurance. In the case of Australia and Canada, however, the warranty limits are expressly stipulated, although the law does not order a warranty deposit. Meanwhile, Japan, China, and Korea are representative countries with a public-led system for home quality and defect repair. The related laws stipulate the warranty deposit as a percentage of the construction cost in these countries. Of course, even in these cases, there is a warranty insurance system in addition to the warranty deposit, and most of them are replaced by warranty insurance rather than directly depositing the warranty deposit. The case of depositing the warranty deposit through a public institution is limited to special cases where the developer's economic credibility is low.

In the UK, a representative private-led system, housing quality assurance has been provided by the National House Building Council (NHBC), an organization of home construction companies [34]. Concerning home defects at the government level, the Housing Defects Act was enacted in 1984 and then incorporated into the Housing Act. It is mainly for securing the quality of a rental home, and it does not stipulate a warranty deposit for defects [35]. In the case of the United States, the relevant home construction law does not require a warranty deposit for repairing defects in a home. Nevertheless, it is common to include a warranty from the developer. The Housing Act of 1954 stipulates that home sellers or builders must provide a home warranty to home buyers [36,37]. Meanwhile, the warranty stipulated in the Magnuson–Moss Warranty Act regulates the warranty for everyday consumable items, and it is regarded as including the home in the consumable items [37,38]. In the United States, home warranty companies such as Home Buyers Warranty (HBW), Residential Warranty Insurance Corporation (RWC), Quality Builder's Warranty Company (QBW), and Professional Warranty Corporation (PWC) are being operated.

In the case of Australia, the warranty for defects was stipulated in Article 102 of the Australian Consumer Law (no.2) in 2010, but the warranty deposit for defect repair was not specified [39]. On the other hand, as in the United Kingdom and the United States, Australia required the developer to purchase home warranty insurance through a home warranty company. According to Victoria's Domestic Building Contracts Act 1995, construction contracts require a warranty, with a deposit of 5% for a contract amount of more than $20,000 and a 10% deposit for contracts of less than $20,000 [40]. In Canada, in the case of the State of British Columbia, the Homeowner Protection Act and Insurance Act stipulates a warranty for a home, but it does not regulate a warranty deposit. Instead, the

following minimum standards are set to establish the limit of the warranty insurance. The ceiling of the warranty insurance for a single-family home must be more than the lesser amount between the owner's purchase price and $200,000. Individual households in a condominium must be more than the lesser amount between the owner's purchase price and $10,000. The entire apartment complex must be more than the smallest among the initial purchase price for the whole, the number of households multiplied by $100,000, and $2.5 million [41].

In the case of Japan, discussions began in 1975 to introduce a warranty system to improve home defects and quality; a detailed plan was disclosed in 1978, and construction companies began to sign up for the warranty in 1980 [42]. On the other hand, the Act on Assurance of Performance of Specified Housing Defect Warranty stipulates that a defect repair deposit of at least JPY 20 million to a maximum of JPY 12 billion shall be deposited depending on the number of households [43]. In the case of China, quality assurance is stipulated according to Article 62 of the Construction Law [44], and according to Article 91 of the Warranty Law, the deposit is to be agreed upon between the contracting parties within 20% of the down payment [45]. In the case of Korea, the Defect Liability System is stipulated in the Housing Act, the Collective Building Act, the Framework Act on the Construction Industry, and the Multi-Unit Housing Management Act to compensate for defects in-home [46–49]. In particular, in the case of a new home warranty under the Housing Act, 3% of the construction cost shall be deposited in cash with a warranty company as a deposit, or a warranty is issued with the condition that the warranty company guarantees within the scope of the deposit and temporarily sends it to the administrative office that is the home licensing authority. Furthermore, it should be re-transferred to the homeowner who moved in after completing the home [46].

*2.2. Defect Repair Cost*

As mentioned above, housing-related laws do not explicitly set a warranty deposit for repairs in private-led countries. In public-led countries, however, the deposit is calculated based on the construction cost ratio or a formula that combines the basic amount with the number of households. Nonetheless, one cannot trace the specific criteria for deciding this. Since the warranty deposit of the home is for repairing the defect, it is necessary to find out how much it costs. Therefore, this section evaluates how much was spent on the repairing cost, using relevant literature.

According to the related literature, quality problems are expressed in various ways, such as repairing defects or failure, rework, and nonconformance. Defect repair is mainly meant to resolve problems after the construction is completed and handed over to the homeowner. Rework means repairing defects identified during construction or works caused by design changes. However, the significant literature also states that it is difficult to distinguish them [31,32]. Therefore, the defect repair cost to supplement the home quality includes what is expressed as defects, failure, rework, and non-conformity in the preceding studies and is based on the repair cost, which is input for the rework against the construction cost or warranty contract amount.

The cost used for quality improvement in a construction project is compared with the construction cost. As can be seen in Table 1, although there are variations depending on the facility that is the subject of research in each preceding study, it is known that, on an average, a minimum of 0.18% to a maximum of 5.4% of the construction cost is required as a repair cost.

The study by Josephson et al. analyzed the cost spent on the rework in the 7 construction projects in Sweden and proposed the rework cost of an average 4.4% as against the construction cost [27]. Among these, home construction was one case, and the repair cost or its ratio to the construction cost was not specified.

**Table 1.** Comparison of DRC ratio to CC in previous studies.

| Researcher | Scope | Case Quantities | | Defect Repair Cost Ratio to Construction Cost | | | Reference |
|---|---|---|---|---|---|---|---|
| | | Total | Home | Minimum | Average | Maximum | |
| Josephson. | Various | 7 | 1 | 2.30% | 4.40% | 9.40% | [27] |
| Mill | Home | Unknown | Unknown | - | 4.00% | - | [28] |
| Hwang | Owner | 181 | Unknown | - | 5.40% | - | [29] |
| | Contractor | 178 | Unknown | - | 2.20% | - | |
| Choi | Home | 48 | 48 | - | 1.10% | - | [30] |
| Forcada | Various | 40 | 1 | - | 2.75% | - | [31] |
| Love | Various | 68 | Unknown | - | 0.18% | - | [32] |
| Liu | Home | 6 | 6 | 2.07% | 4.95% | 10.27% | [33] |

A study by Mill et al. analyzed the ratio of defect repair cost to the guarantee contract amount for home claims in Australia [28]. A total of 10,548 cases received by the Housing Guarantee Fund were reviewed. The minimum was 3.05%, the maximum was 604.1% per year, and the overall arithmetic average was calculated as 4%. Among them, the number of homes and the number of households were not specified, and there was a difference in that they were based on the warranty contract amount, not the construction cost.

A study by Hwang et al. analyzed the rework costs for 359 facilities reported to the construction industry institute (CII) in the United States [29]. The project was divided into cases reported by the developer and the builder, and facilities were subdivided into buildings, heavy industrial facilities, infrastructure, and light industrial facilities. The developers reported 181 cases, and the average cost was 5.4%, of which 32 cases were buildings, which accounted for 4.6% of the total. Builders reported 178 cases, and the average cost was 2.2%; there were 12 cases of construction, and the repair cost was 0%. According to Hwang's study, the repair cost of the developer's case is higher than that of the builder's case, while the defect repair cost of the building construction seems to be relatively low compared to other facilities. However, whether the home was included or how many home projects or numbers were in the research subject building is not explicitly stated.

Choi has conducted a study on 100 home complexes in Korea [30]. The cost of repairing defects in each home complex was based on the total cost determined through litigation. In Korea, litigation is usually conducted twice, around 3 years and just before 10 years of the warranty period. According to Choi's research, it was analyzed that the judgment amount for 48 complexes where lawsuits were filed before or after 3 years of the warranty period was 1.1% of the construction cost. However, the litigation judgment includes the parts that were covered and those that were not covered by the warranty deposit. As described later, the warranty deposit in Korea is assured only for problems that occur after the house is completed and delivered to the buyer. Even if the home is constructed differently from the contractual design drawings or related laws, it cannot be held responsible based on the warranty deposit, and it is interpreted as a separate right.

Forcada et al. conducted a comparative analysis of 40 construction projects in Spain according to project type, region, subject, and method. According to the results, it was analyzed that the average rework cost was 2.75% [31]. In the case of buildings, the rework cost was higher than that of civil engineering, and the international project had more rework costs than the local domestic project. In addition, it was found that private projects had higher rework costs than public projects, and joint projects had higher rework costs than single projects. However, civil engineering was three times more than buildings, and only one home construction project was included.

A study by Love et al. analyzed rework data for 218 projects from Australian construction companies [32]. Cases were classified into civil, building, power, rail, heavy industry, water, and telecommunication, and analysis results for 68 projects with general construction costs showed that the rework cost was 0.18% on average compared to the construction cost. The heavy industry project had the highest rework cost ratio, and the water project

had the lowest rework cost ratio. The rework cost ratio of the building project was 0.02%, the second-lowest after the water project. However, it is not certain whether the building included homes in the case.

A study by Liu et al. analyzed the rework cost for 6 residential buildings in China [33]. A minimum of 2.07% and a maximum of 10.27% were required for rework compared to the construction cost, and the overall average was analyzed to be 4.95%.

Preceding studies reviewed various facilities' related defect repair costs and rework costs. The ratio of defect repair cost to construction cost varied from case to case, and the difference between the most minor case and the most significant case was 30 times, even on average. In addition, there were few studies conducted on homes, and the number of subject cases was small. Therefore, it is necessary to limit the subject to homes and collect and analyze sufficient cases to determine and establish the standard of the warranty deposit using the home defect repair cost.

*2.3. Project Conditions*

Since comparing the home defect repair cost and the warranty deposit can be viewed as an ex-post quality evaluation of the home, whether there is an effect by conditions such as the execution and construction of the home construction project will be discussed.

Are there differences in drive method, construction cost, quality, etc., for each public and private sector in project implementation and construction? Is it better for the public or better for private interests? It is difficult to determine the answer to such a question. In general, generating a profit is not a goal in the public sector [50], but the private sector should make a profit [51]. On the one hand, the construction and supply of affordable homes is an important policy goal in many countries, regardless of whether it is profitable or not [52]. Therefore, even in the public sector, wasting the budget leads to social loss, so more homes should be built within the same budget as much as possible [53]. In addition, the purpose of the public existence is to provide homes that can maintain an appropriate level of quality, even though it is not the highest quality. Of course, other opinions also exist in which there is no difference in quality between the homes built by the public sector and those by the private sector [54]. A study by Forcada et al. discussed earlier also suggested a difference between the case where the developer was public and the case where the developer was private [31]. Therefore, this study will examine whether there is a difference in home quality depending on the project implementor (i.e., public or private).

There can be differences in the drive method, construction cost, and quality aspects depending on the developer who conducts the production, whether it be a landlord project, trust project, or joint project (where multiple entities participate). A trust project is advantageous in terms of financing and efficiency, has high investment value by achieving economies of scale, and is excellent in terms of operational efficiency and tax deduction [55–57]. On the other hand, some argue that the trust project is disadvantageous. Representative opinions are that the trust project did not achieve economies of scale and that the rate of return is high, but the risk is also high, making the real estate market unstable [58–60]. However, these studies fail to provide opinions on the superiority or disadvantage of the landlord project compared to the trust project in terms of quality. Therefore, in this study, one may examine whether there is a difference in quality depending on the composition method of the participants in the execution of the home construction project.

The project execution mode can be divided into direct management (owner project, OP), a case in which the developer directly constructs the house, and a contract project (CP), which is a case of requesting the construction from a specialized company. Depending on who will perform the home construction, there is a difference in drive method, construction cost, quality, etc. The performance of a construction project depends on the will and responsibility of the developer [61]. As a decision-maker on key issues, the developer manages and provides information on the achievement of project goals and performance, and there is an opinion that the will and dedication of the developer lead to improvement

in project performance [62]. However, in the general case, the developer is ignorant of construction, so he hardly understands the details and does not know how to manage the project [63] efficiently. Lack of expertise in the construction project causes problems such as increased construction cost and a delay in construction lead time, leading to project failure [64]. For this reason, it is crucial to select an efficient procurement method and a good builder. In addition, it is common for the developer not to directly perform the construction (delivery) in the construction project but to entrust it to a separate specialized company. Therefore, in this study, the difference in quality is determined, depending on whether the house construction is directly managed or contracted to the builder.

There can be a difference between a method in which a single entity plays the role of a builder who is in charge of the entire home construction (Single Prime Contract; SPC) or a method in which a separate contract for each specialized field and construction management is performed (multiple prime contract; MPC). In general, it is considered that separate subcontracting (MPC) for each work type is advantageous for companies in a specialized field and disadvantageous for existing single subcontracting (SPC). Additionally, one of the reasons for denying or opposing separate subcontracting is that it causes waste due to rework [65,66]. However, according to a study conducted by Debella and Ries on school construction projects, there was no difference in unit cost, construction speed, cost growth, change orders, and project delays between separate and single contracts [67]. Therefore, this study will also compare whether there is a quality difference between single prime and multiple prime subcontracts in home construction.

*2.4. Lawsuit Issues*

The home warranty deposit is for the case where the developer becomes insolvent and cannot properly repair the defect. Therefore, it is necessary to examine whether there is a difference in the cost of repairing defects according to the incapacity of the project entity. If there is a difference, it is necessary to devise complementary measures.

There are always concerns if the developer becomes insolvent, such as bankruptcy or court management [20–26]. However, it is unknown how many cases there are in which the developer has become insolvent in the home construction project and what effect the repair of defects due to the developer's incapacity has had. Therefore, it is necessary to determine the extent to which the project entity is insolvent and whether there is a difference in the repair cost where the business entity is normal and where it is insolvent.

Meanwhile, depending on the outcome of the litigation, the judgment amount is jointly shared among the parties involved in the litigation. It is necessary to find out how each party shares the judgment amount. The judgment of the defect litigation is mixed with claims for damages instead of repairs in the case of direct claims for warranty deposits. Usually, if the developer is not in an insolvent condition, the warranty company will not pay the deposit, but in some cases, some deposits are paid depending on the interpretation of the legal relationship. In addition, there are cases where all project parties are insolvent, and the warranty company has to bear all the costs. Therefore, it is necessary to determine whether there is a difference according to these cases.

## 3. Materials and Methods

*3.1. Object and Scope*

In this study, a home defect refers to a problem that interferes with a home's safety, function, and aesthetics from construction till use. Defect repair refers to an action to remedy these problems that occur in the home. The cost involved in the defect repair is defined as the defect repair cost. The home warranty deposit is provided if the developer or builder fails to perform repairs due to incapacity. The home warranty deposit refers to an amount that is the limit of warranty when depositing cash with a public institution or insurance company or purchasing insurance.

The study was conducted by targeting homes in Korea. According to the Korea Housing Act, the defect repair deposit is 3% of the construction cost. In addition, defects

that fall within the scope of the home warranty deposit are limited to defects that occur after the home construction is completed and handed over to the owner (defects that occurred after delivery; post-handover defects). Most of the post-handover defects include common problems such as cracks and leaks. On the other hand, pre-handover defects are not covered by the home warranty deposit. Therefore, in this study, the repair cost was limited to the defects that occurred after handover, and the repair cost for the defects that occurred before delivery was excluded.

This study deals with the cases where litigation was made for home defects in Korea. When a lawsuit is filed, an expert (appraiser) appointed by the court investigates the defect. The appraiser calculates the cost of repairing defects based on the standards set by the court. It is calculated according to the Construction Appraisal Practice [68], a guideline for judging whether there is a defect, and the construction work standard estimating system [69] for calculating input manpower for the work.

### 3.2. Data Collection

To conduct this study, 474 cases of lawsuits filed in Korea between 2007 and 2019 were collected. Each case contained at least one housing complex. The judicial precedent contains information such as defect repair costs and deposits confirmed by the defect litigation. The repair cost for defects that occurred before handover and those that occurred after handover are separately listed in the judicial precedent. As mentioned in Section 3.1 above, only defects after handover are protected by the home warranty deposit in Korea. Therefore, only repair cost information for defects after handover was extracted from the home warranty deposits in the precedent. Deposit information is also described in the judicial precedent, but not all cases dispute the home warranty deposit. If it is judged that the developer or builder is not legally insolvent, there is little room for a deposit to be paid. Therefore, the home warranty deposit is claimed only when the developer or builder is assumed to be insolvent. Among the 474 collected cases, 290 were identified in which the deposit was specified. Therefore, the analysis was conducted for these 290 cases.

On the other hand, the construction cost was not specified in the judicial precedent. However, the construction cost could be calculated if the deposit is specified in the judicial precedent. The home warranty deposit of 3% of the construction cost is stipulated in Korean law. Therefore, the cost of building a home could be calculated as the deposit divided by 3%. As shown in Figure 1, the analysis subject is 290 home complexes, with 309,422 households sold as dwelling houses. As the distribution by region, Seoul, Incheon, and Gyeonggi Province, the capital areas, had 154 cases and 198,921 homes, accounting for about 50% of the total. Gangwon-do had 10 cases and 8,350 households, and Chungcheong-do, including Chungbuk, Chungnam, Sejong, and Daejeon, had 37 cases and 26,859 households. Jeonbuk, Jeonnam, and Gwangju had 17 cases and 12,015 households, while Gyeongbuk, Gyeongnam, Daegu, Ulsan, and Busan had 71 cases and 63,277 households. By applying the exchange rate calculated from 1 USD to 1,179 Korean won (KRW), the total amount of deposits in all cases amounts to 1.253 trillion KRW and 1.06 billion USD. In addition, the total cost of repairs for defects after use and inspection amounts to KRW 184 billion and 156 million USD.

### 3.3. Comparison Method

In this study, to find out the appropriate level for the home warranty deposit, the defect repair costs determined in the case of litigation were compared. As reviewed in the literature review, the countries that stipulated the warranty deposit for home defects in their laws were Japan, Korea, and China. According to the Japanese standard, the number of households is proportionally applied to the base amount set in Japanese yen, and it is added in proportion to the construction cost in Korea and China. However, since the case of this study is focused on Korean homes, it is not easy to compare them based on the basic amount set by the Japanese standards. Because this basic amount seems to have been established in consideration of the situation in Japan, it is challenging to apply it

directly to the case of Korea. Therefore, it is necessary to reset the basic amount according to the circumstances of the case, considering the difference in house prices and exchange rates, which is beyond the main scope of this study, so it might be reasonable to conduct a separate study in the future to apply this method. In addition, the Chinese standard is set to within 20% of the construction cost, which is too high compared to the average level, enough not to be used as a comparison standard. Therefore, it is compared with the Korean standard, which sets the home warranty deposit at 3% of the construction cost.

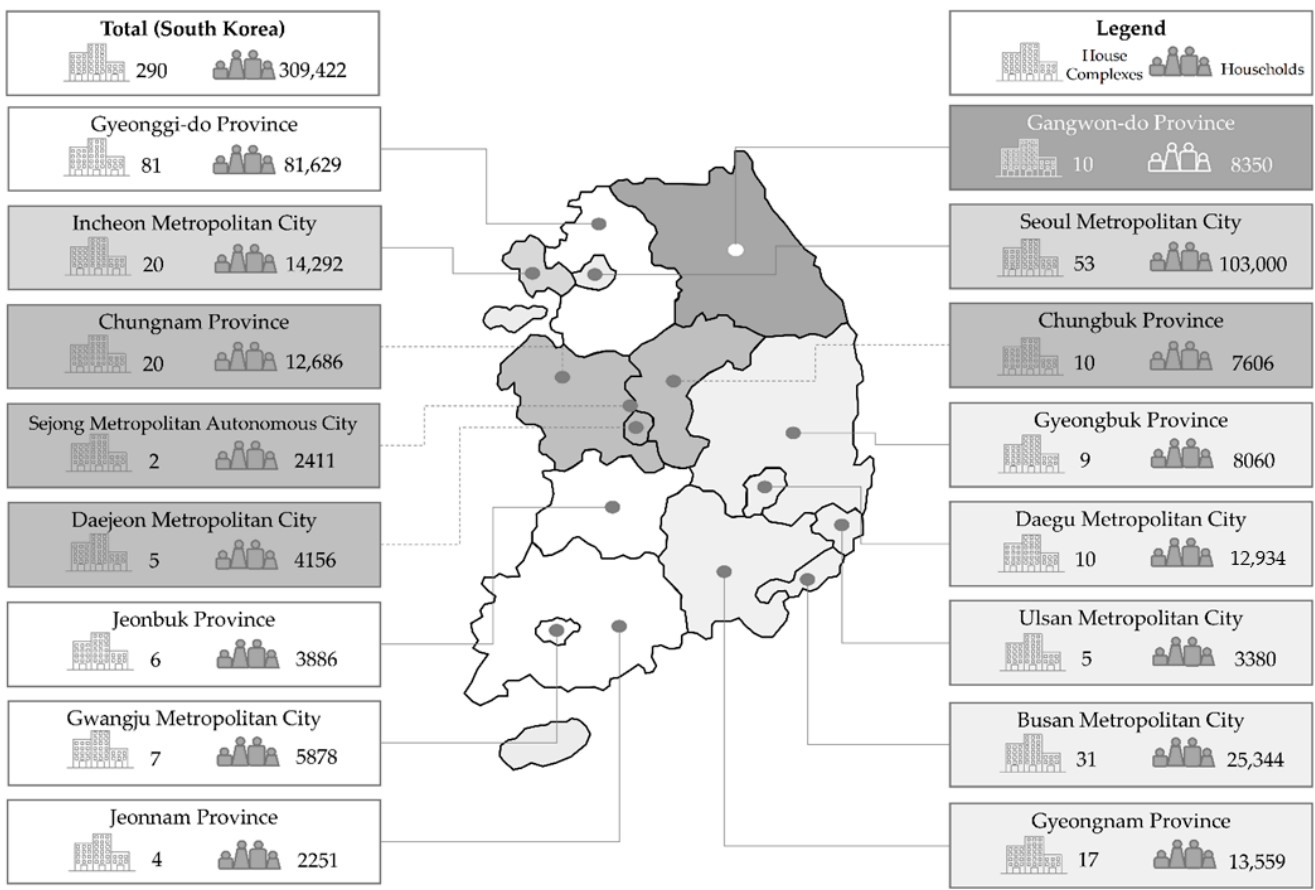

**Figure 1.** Distribution of study case.

This study's purpose was to determine whether the standard of the home warranty deposit is appropriate. If the cost of repairing a home requires more money than the warranty deposit, the deposit standard should be revised. On the other hand, if the repair cost is less than the deposit, it would be a social waste to provide for an excessive deposit. Therefore, the defect repair cost and the deposit confirmed in the research cases were compared. It was possible to compare the repair cost and the deposit directly. However, as mentioned in Section 3.2 above, these amounts were significant, at least in billions and trillions, so it was not easy to judge intuitively. On the other hand, as mentioned in Sections 2.1.2 and 2.2, defect repair costs were evaluated as a ratio to the construction cost in most cases. Therefore, the ratio of the defect repair cost to the housing construction cost and the home warranty deposit ratio confirmed in the research case were compared in this study.

In this study, construction cost (CC) referred only to the cost invested in constructing a home. Therefore, the construction cost included structure, equipment, finishing, and landscaping. The plot prices were excluded. A home warranty deposit (HWD) is set at 3% of construction cost (CC), as shown in Equation (1). Therefore, the construction cost was calculated in reverse from the defect repair deposit. The ratio of defect repair cost to

home construction cost (DRC ratio) was calculated as a percentage obtained by dividing the defect repair cost by the construction cost, as shown in Equation (2).

$$\text{Home warranty deposit} = \text{construction cost} \times 3\%, \tag{1}$$

$$\text{Defect repair cost ratio} = \text{defect repair cost} \div \text{construction cost} \times 100 \ (\%), \tag{2}$$

The analysis has proceeded as follows. First, it was examined whether the ratio of the defect repair cost to construction cost for 290 cases exceeded the 3% prescribed by law. If the defect repair cost to construction cost ratio exceeded 3%, how much it exceeded and what the difference was between cases exceeding 3% and cases under 3% were additionally reviewed. However, in this study described later in Section 4.1, there was no case in which the defect repair cost to construction cost exceeded 3%.

Second, the case where the ratio of defect repair cost to construction cost was less than 3% was considered. In this case, the average, maximum, and minimum defect repair cost ratio of 290 cases was found, and their distribution pattern within the subdivided section was within 3%. Moreover, the appropriate level was sought if it was adjusted downward from the current standard of 3%.

Third, since this study compared construction costs, the difference in the defect repair cost ratio according to the construction cost scale was considered whether the scale of construction cost increased; the defect repair cost ratio also increased; or, conversely, it tended to decrease. A group that might have be different in terms of the defect repair cost ratio by construction cost scale was set, and an ANOVA was performed on each group's defect repair cost ratio. When the ANOVA result confirmed a significant difference in the average defect repair cost ratio between groups, and the particulars were derived from the composition of each group's defect repair cost ratio, the defect repair cost ratio for each group was proposed based on this.

Additionally, whether there was a difference in the defect repair cost ratio according to project conditions or litigation issues and whether there was a significant difference between the defect repair cost ratios for each subgroup according to each issue were checked. A *t*-test or ANOVA was selected and analyzed according to the number of subgroups for each issue, and IBM SPSS Statistics ver. 21 was used.

## 4. Results

### 4.1. Defect Repair Cost and Home Warranty Deposit

4.1.1. Comparison of Total Case

In order to find out whether the standard for calculating the home warranty deposit (HWD), which is set at 3% of the construction cost, is appropriate, 290 home complex cases were analyzed. This is to determine whether the defect repair cost (DRC) required to repair the defect exceeds the warranty deposit standard and, if so, to ascertain what level is reasonable. Conversely, if the defect repair cost (DRC) required is less than the standard for the deposit, how much it would be reasonable to adjust was investigated. In addition, this study assessed if the deposit standard needs to be raised or lowered in each case and what level would be appropriate.

The results say that there was no case where each case's defect repair cost ratio exceeded the warranty deposit ratio. Figure 2 shows the ratio of the home warranty deposit and the defect repair cost to the construction cost for each case. The home warranty deposit is fixed at a rate of 3% of the construction cost (CC) according to the standards of the Act. On the other hand, the ratio of defect repair cost to construction cost is different because it is a percentage value obtained by dividing the construction cost by the defect repair cost for each case. Notably, it was confirmed that there was no case where the ratio of the defect repair cost to the construction cost exceeded the home warranty deposit ratio. The average defect repair cost ratio was 0.538%, the minimum value was 0%, and the maximum value was 2.22%. There were five cases where the defect repair cost ratio exceeded 1.5%, and two cases showed that the defect repair cost ratio exceeded 2%.

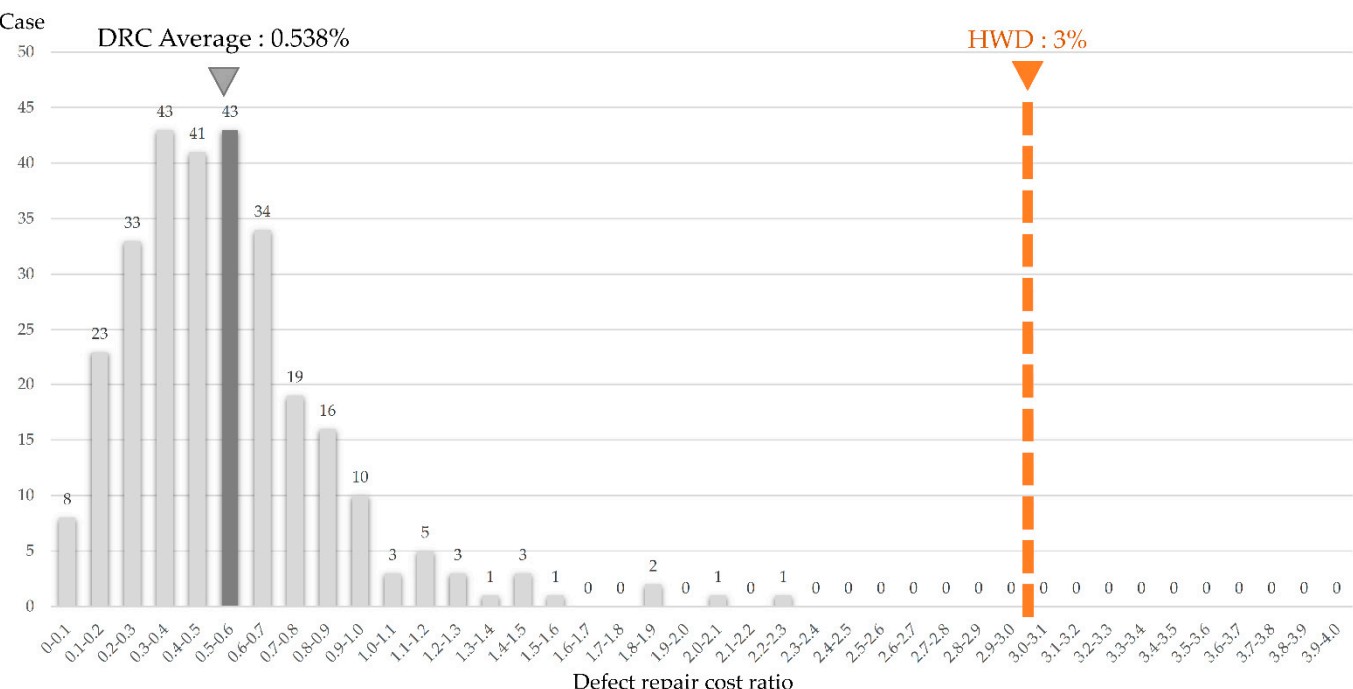

**Figure 2.** Comparison of DRC ratio and HWB ratio.

When this analysis result was compared with the analysis results of previous studies listed in Table 1 above, the average value was found to be about an intermediate value between Love's study [32] and Choi's study [30], and the rest of the studies suggested that there is a big difference compared to the one result value. Even based on the maximum value of the repair cost of 2.22% in this study, it is slightly higher than the minimum value of 2.07% suggested in Liu's study and even lower than the minimum value of 2.30% suggested in Josephson's study [27].

Additionally, compared with studies that analyzed only homes, Mills [28] presented 4.00% and Liu [33] 4.95% as the average value. The result of this study, being 0.538%, which is 1/8 of the above level, seems very low. Compared with the average of 1.10%, which is the study result of Choi [30], the defect repair cost ratio in this study case was about half, which is a significantly lower value.

According to the previous review, there were no cases in which the defect repair cost exceeded the warranty deposit, so reducing the ratio in Korea seems reasonable, which sets the warranty deposit at 3% of the construction cost. However, it is necessary to examine what level is reasonable to adjust to. As mentioned above, since the current standard for home warranty deposit in Korea is based on the construction cost, it was additionally analyzed to see if there was a difference depending on the construction cost scale.

4.1.2. Comparison of The Construction Cost Scale

Figure 3 shows the defect repair cost and home warranty deposit for each case according to the scale of the construction cost. The *X*-axis is the construction cost, the *Y*-axis is the defect repair cost, and the warranty deposit is 3% of the construction cost, so it is distributed linearly. As shown in Figure 2, the repair cost is within the range of the warranty deposit in all cases. Therefore, the repair cost is always distributed below the deposit distribution. The defect repair cost also increases with the increase in construction cost, but it is difficult to conclude that it simply increases proportionally.

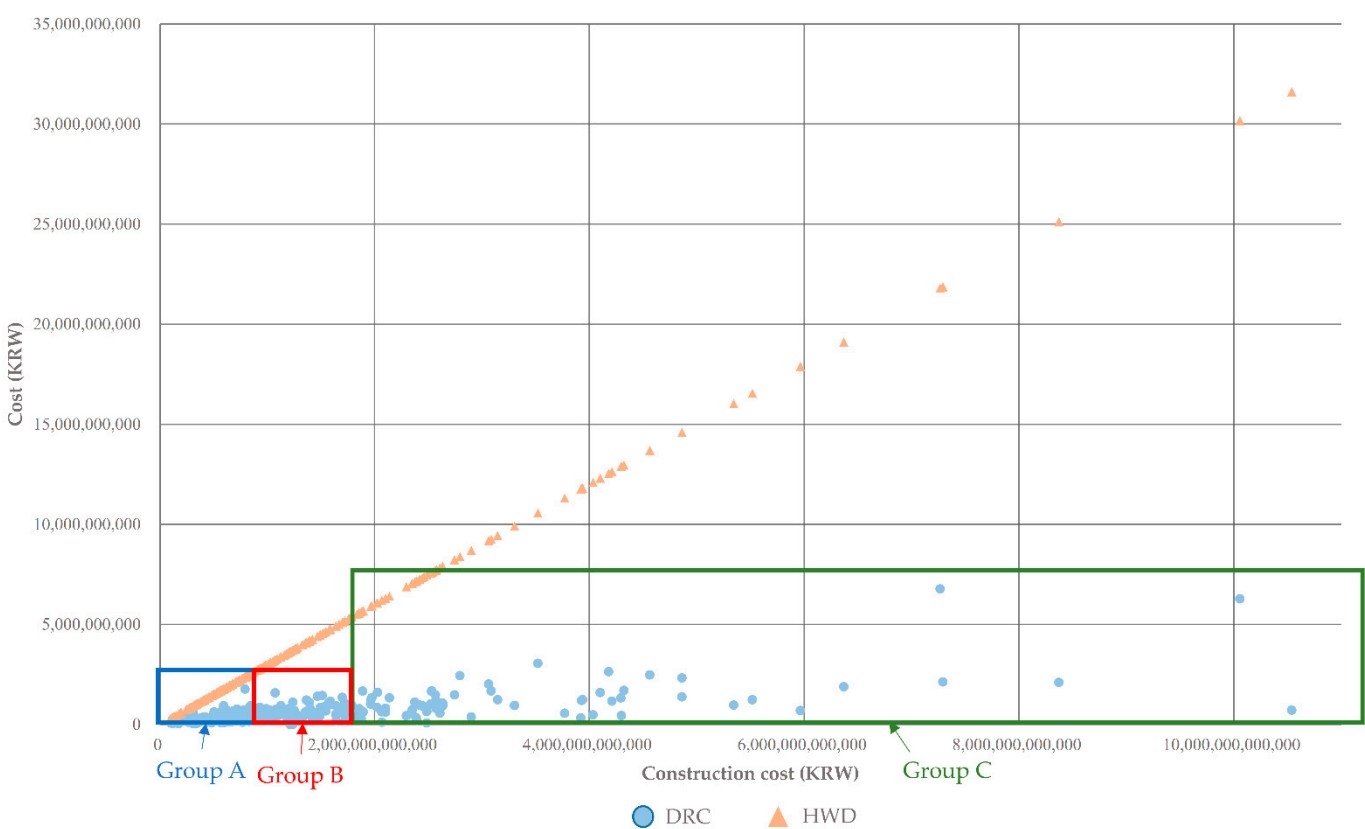

**Figure 3.** Distribution of DRC and HWD to CC scale.

In order to find out the trend of defect repair cost according to the construction cost in detail, cases of projects were divided into three groups: Group A for cases less than 800 billion KRW, Group B for cases between 800 billion KRW and 1.6 trillion KRW, and Group C for cases of 1.6 trillion KRW or more. In addition, the distribution of the ratio of defect repair cost to construction cost was reviewed for each group. The distribution was examined by setting the interval from the minimum 0% to the maximum 3%, the warranty deposit standard, in 0.5% increments each.

To find out the trend of defect repair cost according to the construction cost in detail, the distributions of the construction cost size were compared. The construction cost was categorized into KRW 500 billion units and KRW 1 trillion units, but no significant difference was found. However, there was a significant difference in the defect cost by the construction cost in the classification of KRW 800 billion and KRW 1.6 trillion units. Therefore, these cost units were adopted to compare the defect repair cost with the construction cost.

For Group A, the minimum construction cost was 98.8 billion KRW, and the maximum was 798.5 billion KRW. As shown in Figure 4, the defect repair cost ratio to Group A's construction cost was an average of 0.67%, a minimum of 0.09%, and a maximum of 2.22%. For Group B, the minimum construction cost was 80.6 trillion KRW, and the maximum was 1.59 trillion KRW. As shown in Figure 4, the ratio of defect repair cost to construction cost of Group B was 0.50% on average, 0% at the minimum, and 1.47% at the maximum. For Group C, the minimum construction cost was 1.64 trillion KRW, and the maximum was 10.54 trillion KRW. As shown in Figure 4, the ratio of defect repair cost to construction cost in Group C was 0.39% on average, 0.03% at the minimum, and 0.93% at the maximum.

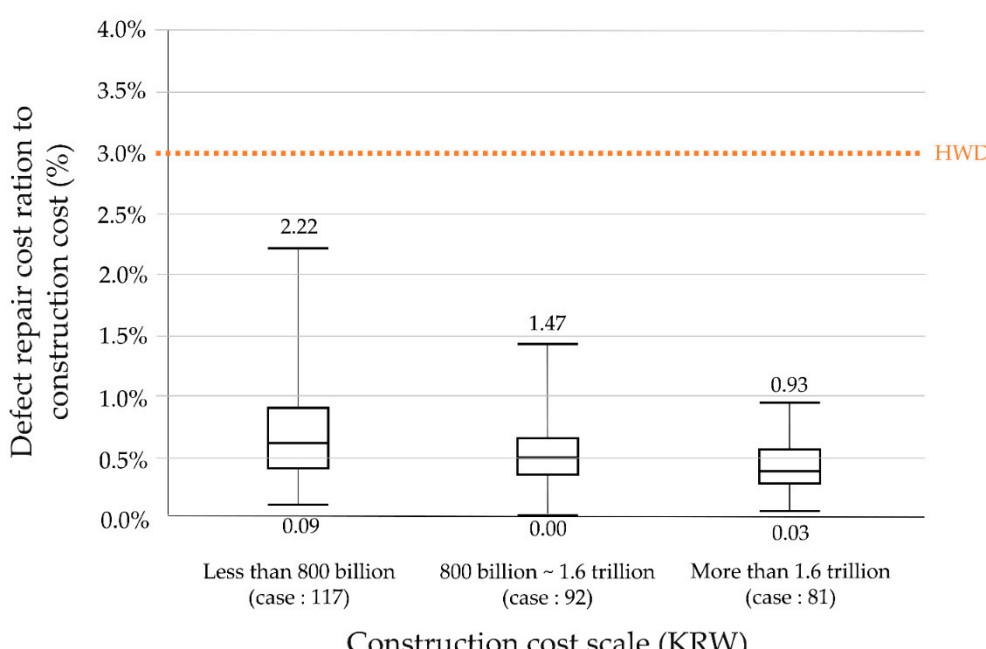

**Figure 4.** Comparison of the construction cost scale.

ANOVA analysis was performed to check whether there was a difference in the defect repair cost ratio for these three groups. The analysis standard was based on the average value. The null hypothesis stated that there was a difference in the defect repair cost ratio for each case, and the alternative hypothesis indicated that there was no difference. Appendix A, Figure A1, the result of ANOVA analysis, shows that the sig value of ANOVA was 0.000, which was less than 0.05, so the null hypothesis was accepted. Therefore, it is clear that the average defect repair cost ratio between the three groups according to the construction cost scale is different. The average defect repair cost ratio of each group was 0.67% for Group A, 0.50% for Group B, and 0.39% for Group C, and it could be regarded that the cases belonging to the group with a larger construction cost had a lower defect repair cost ratio.

Figure 5 shows the defect repair cost ratio distribution for each group divided by total cases and construction cost scale. There were 270 cases where the defect repair cost ratio of all cases was less than 1% of the construction cost, which was 93.10%. There were 15 cases between 1% and 1.5%, 3 cases between 1.5 and 2%, and 2 cases between 2% and 2.5%. In the case of Group A, where the housing construction cost was less than 80 billion won, there were 99 cases in which the defect repair cost ratio was less than 1% of the construction cost, accounting for about 84.6% of Group A. On the other hand, there were 18 cases exceeding 1%, accounting for 15.4%; 3 cases exceeding 1.5%; and 2 cases exceeding 2%. All cases in which the ratio of defect repair cost to construction cost exceeded 1.5% among all study cases belonged to Group A. In the case of Group B, where home construction costs ranged from 80 billion KRW to 160 billion KRW, there were 90 cases in which the repair cost was less than 1% of the construction cost, accounting for 97.8% of Group B.

On the other hand, there were only two cases where the repair cost exceeded 1%, and none of the cases exceeded the repair cost of 1.5%. In all cases belonging to Group C, where the housing construction cost was 160 billion KRW or more, the ratio of the defect repair cost to the construction cost was less than 1%, so there was no case where it exceeded 1%. As reviewed above, the defect repair cost ratio was higher on average in complexes with a small construction cost, and there were cases where it exceeded 1.5%.

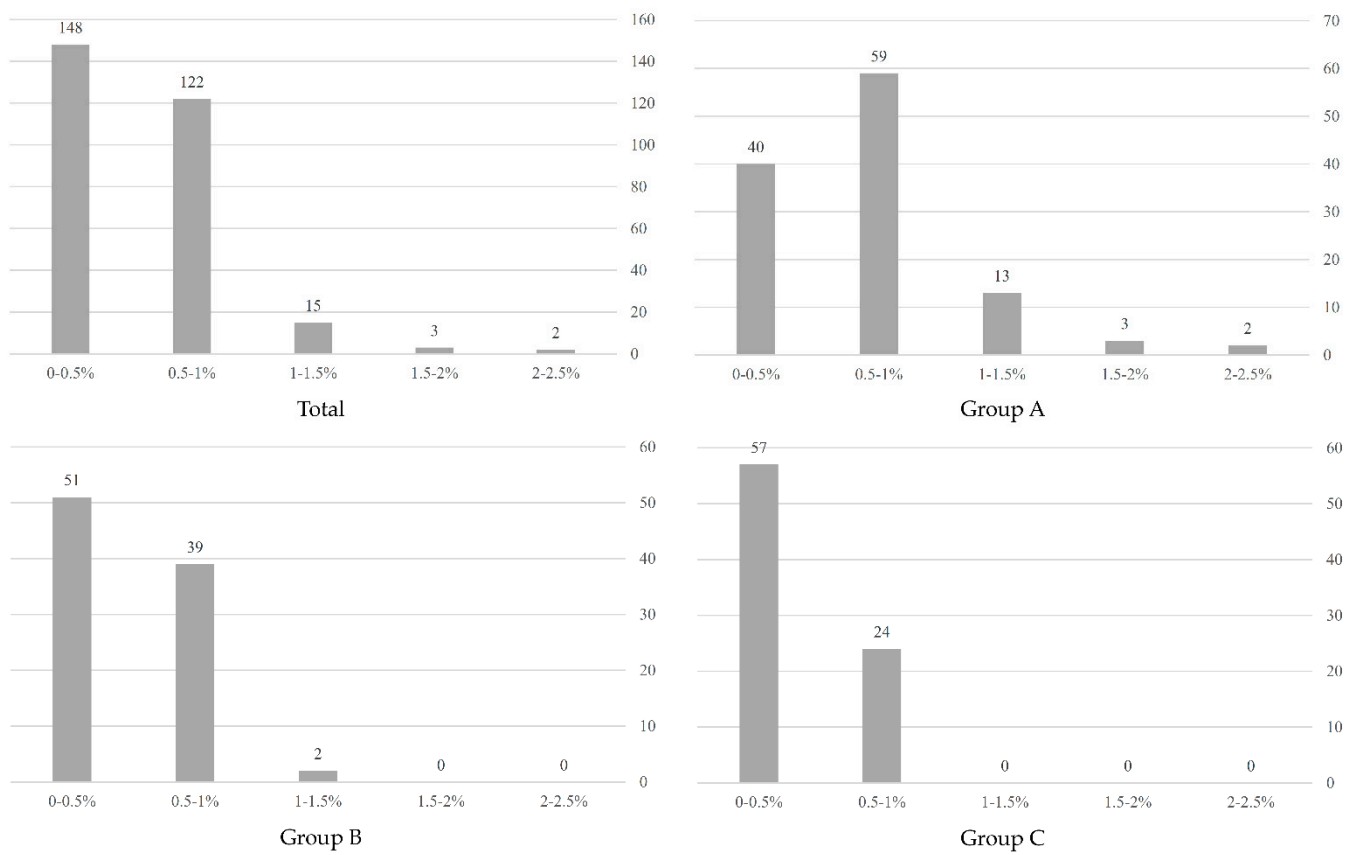

**Figure 5.** Histograms of groups to construction cost scale.

*4.2. Comparison of Project Conditions*

4.2.1. Project Execution Sector

In the case of private, public, and joint home construction projects, it was investigated whether there was a difference in the ratio of defect repair cost to construction cost. Among the cases, the private project execution accounted for the majority, and the joint project execution accounted for very few. In the case of the defect repair cost ratio in each case, it seemed that the case of private execution was somewhat lower than the public execution. Additionally, according to the *t*-test on the average defect repair cost ratio, there was a statistically significant difference between public and private execution.

As shown in Figure 6, there were 264 cases of private execution, with a defect repair cost ratio of 0.525%, but 23 cases were public execution, with a defect repair cost ratio of 0.672%, and 3 cases of joint execution, with a defect repair cost ratio of 0.633%. ANOVA analysis results showed that the number of jointly executed projects was too small to be analyzed as significant. Accordingly, the *t*-test was conducted only for private and public executions, excluding joint execution. The null hypothesis stated that there was a difference in the defect repair cost ratio in the case of private execution and joint execution, and the alternative hypothesis established that there was no difference. From the resulting Appendix B, Figure A2, since Levene's sig value was 0.635, which was more than 0.05, it could be considered that dispersion homogeneity was assumed. In this case, since the sig value of the *t*-test was 0.042, which was less than 0.05, the null hypothesis was adopted. Thus, there was a difference in the defect repair cost ratio between the private and public sectors. Therefore, since the ratio of repair costs was 0.525% for the private sector and 0.672% for the public sector, the repair cost ratio was lower in the private sector than in the public sector.

Among the preceding major studies reviewed above, Martin and Westerhoff suggested no difference in quality between the public sector and the private sector [54]. Additionally,

Forcada et al. suggested that the public sector had lower rework costs than the private sector [31]. However, the analysis results of this study showed that the ratio of defect repair costs in the public sector was higher than that of the private sector in the Korean Housing Construction Corporation.

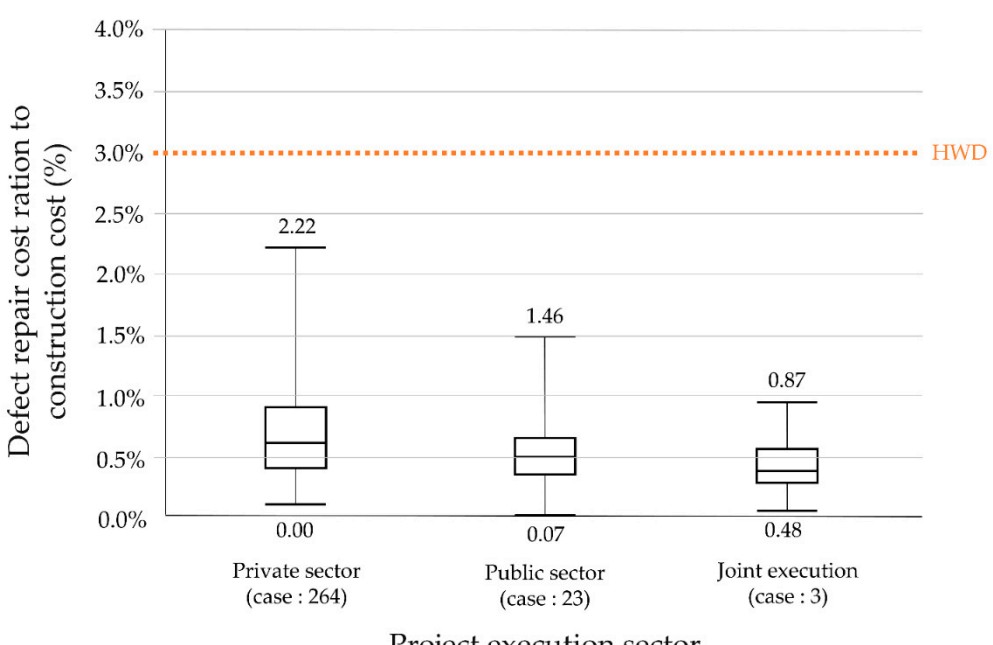

**Figure 6.** Comparison of project execution sector.

4.2.2. Project Implementation Type

The project implementation method can be divided into a case in which a trusted company implements the project, a case in which the landlord directly implements the project, and a standard method in which various entities participate. In each case, the ratio of defect repair cost to construction cost showed only a slight difference. ANOVA was performed to determine whether these differences were significant. However, there was no statistically significant difference in the ANOVA results. Most of the cases were implemented by the landlord alone, and there were relatively few trust projects or joint projects. Moreover, it was found that there was no difference in the defect repair cost ratio between the project execution methods.

As shown in Figure 7, out of 290 cases, there were 20 cases where the landlord let the trust company execute the project, and the defect repair cost ratio was 0.528%. There were 244 cases where the landlord executed the project by himself, while the defect repair cost ratio was 0.535%; 26 cases were jointly implemented projects, and the defect repair cost ratio was 0.574%. The null hypothesis demonstrated a difference in the defect repair cost ratio for each case in the ANOVA analysis, while the alternative hypothesis showed no difference. As shown in Figure A3, the ANOVA analysis result, since Levene's sig value was 0.956, was more significant than 0.05, and dispersion homogeneity was assumed. In addition, since the sig value of ANOVA was 0.840, which was more than 0.05, the null hypothesis was rejected. Therefore, according to the project implementation method, there was no difference in the defect repair cost ratio.

In this regard, among the preceding studies, Cotter and Richard [55], Topuz and Isik [56], and Grybauskas and Pilinkiene [57] emphasized the merits of trust implementation. On the contrary, Ambrose et al. [58], Vogel [59], and Kawaguchi et al. [60] presented the view that trust implementation was disadvantageous. However, according to the previous analysis results, there was no difference in the defect repair cost ratio according to the project implementation method. Therefore, it is not easy to assert a difference in quality

depending on the project implementation method nor to claim that trust implementation is superior in quality compared to other cases.

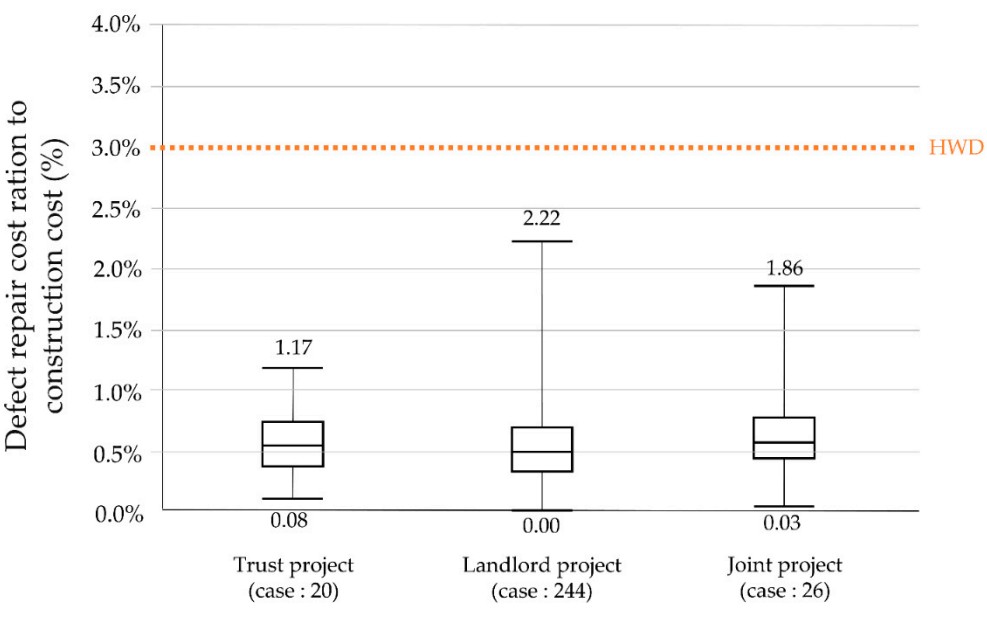

**Figure 7.** Comparison of project implementation type.

### 4.2.3. Building Execution

This study compared whether there was a difference in the ratio of defect repair costs depending on who was the project owner of the home construction. The number of cases of the contract project was three times higher than that of the owner project. According to the results of comparing the defect repair cost ratio between the case of the contract project and the case of the owner project, the case of the contract project was slightly lower, but it was difficult to say that there was a significant difference. Additionally, there was no notable difference in the results of the *t*-test on the average defect repair cost ratio.

As shown in Figure 8, there were 221 cases with a defect repair cost ratio of 0.523%, but 69 cases of directly managed construction had a defect repair cost ratio of 0.585. In the *t*-test analysis of these, the null hypothesis indicated a difference in the defect repair cost ratio in the case of the contract project and direct management construction, whereas the alternative hypothesis showed no difference. As shown in Figure A4, since Levene's sig value was 0.052 or more, dispersion homogeneity was assumed. In this case, the sig value of the *t*-test was 0.182, which was less than 0.05, so the null hypothesis was rejected. Therefore, there was no significant difference between the contract project and the owner project.

The viewpoints of preceding studies were divided into Zou [61] and Korkmaz et al. [62], who emphasized the advantages of the owner project, and Neap and Aysal [63] and Serpell [64], who conversely emphasized the advantages of the contract method. In addition, a study by Hwang et al. suggested that the owner project method had a higher defect repair cost ratio than the contract method [29]. According to the analysis results of this study, the contract method had a slightly lower defect repair cost ratio than the owner project method, but the difference between the two was not statistically significant. Therefore, it is difficult to say whether the quality of the owner project method is significantly lower than that of the contract method.

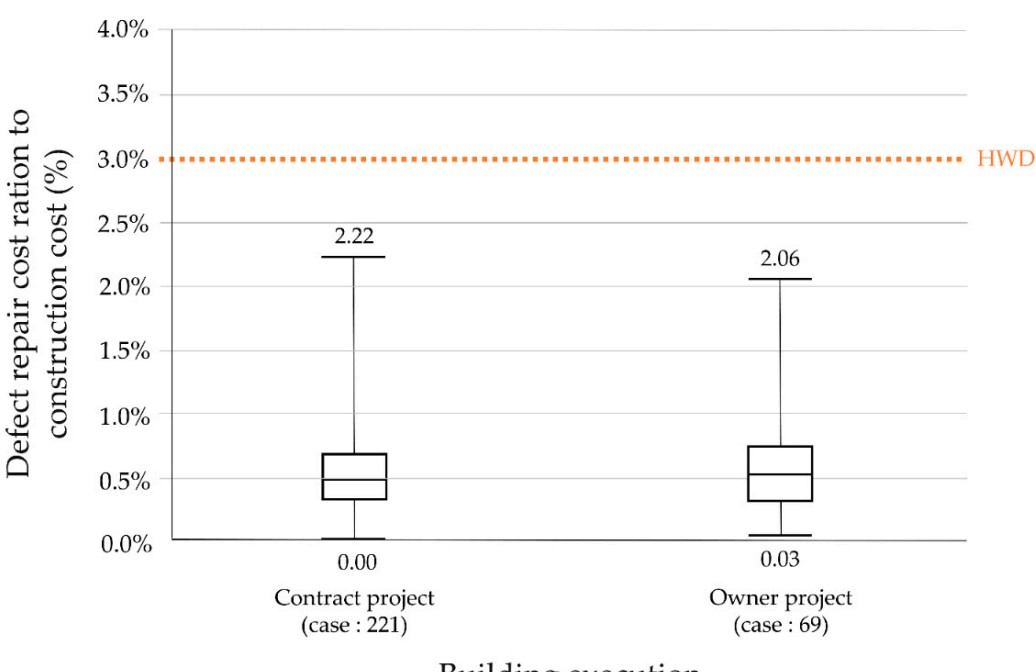

**Figure 8.** Comparison of building execution.

### 4.2.4. Building Contract

This study compared whether there is a difference in the defect repair cost ratio between the case of a single prime contract being made by a single entity and the case of a separate order being constructed by multiple specialized companies (multiple primes). Single prime contracts were twice as numerous as multiple primes, and the ratio of defect repair costs was lower for single prime contracts than for multiple primes. Additionally, according to the *t*-test on the average defect repair cost ratio, there was a statistically significant difference between single prime contracts and multiple prime contracts.

As shown in Figure 9, in the case of a single prime, there were 192 cases, and the defect repair cost ratio was 0.508%, but in the case of multiple primes, the defect repair cost ratio was 0.596%, with 98 cases. In the *t*-test analysis of these, the null hypothesis indicated a difference in the defect repair cost ratio in the case of single prime and multiple primes, while the alternative hypothesis showed no difference. As shown in Figure A5, since Levene's sig value was 0.018, which was less than 0.05, dispersion homogeneity was not assumed. In this case, the sig value of the *t*-test was 0.048, which was less than 0.05, so the null hypothesis was adopted. There was a statistically significant difference between a single prime contract and multiple prime contracts. Therefore, the defect repair cost ratio was 0.508% for the single prime contract and 0.596% for multiple prime contracts, so it could be regarded that the defect repair cost ratio was lower for the single prime contract than for multiple prime contracts.

Nooteboom [65] and Al-Hammad [66] suggested that a single prime contract was more advantageous than multiple prime contracts, whereas Debella and Ries [67] argued that there was no difference between the two. According to the above analysis result, a single prime contract's defect repair cost ratio was lower than a multiple prime contract's. Therefore, the results of this study agreed with Nooteboom [65] and Al-Hammad [66].

### 4.3. Comparison of Lawsuit issues

### 4.3.1. Developer's Insolvency

It was examined whether there was a difference in the ratio of defect repair costs depending on the capability of the developers of the home construction project. There were more cases in which the developer was in a normal state than in the case of insolvency.

In addition, there was almost no difference in the defect repair cost ratio between them. There was no notable difference in the results of the *t*-test on the average defect repair cost ratio too.

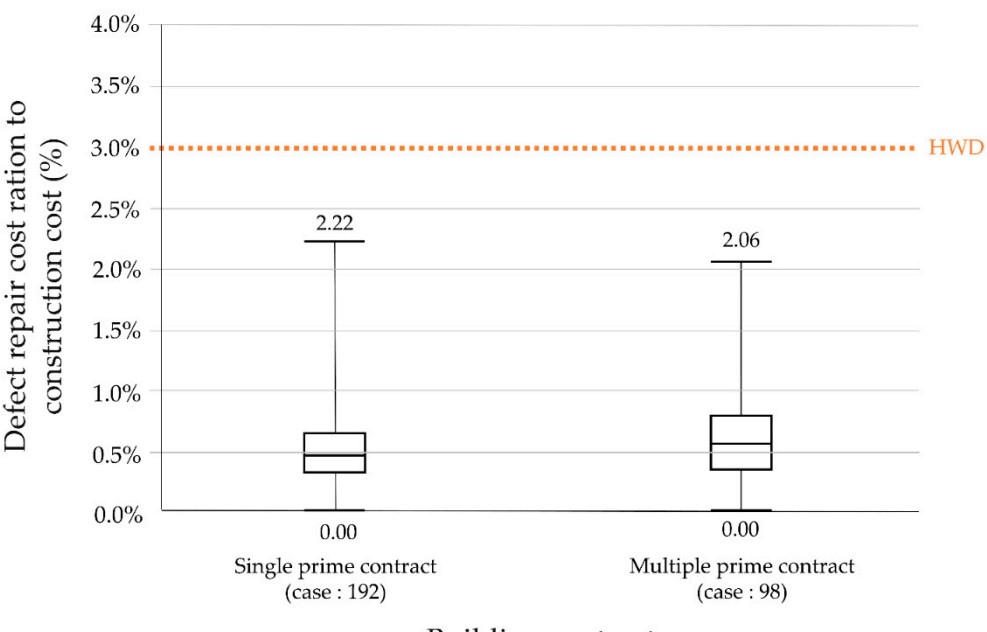

**Figure 9.** Comparison of building contract.

As shown in Figure 10, the defect repair cost ratio was 0.538% in 185 cases in which the developer was in a suitable operation condition, but in 105 cases in which the developer was insolvent, the defect repair cost ratio was 0.539%. In the *t*-test analysis of these, the null hypothesis was set as a difference in the defect repair cost ratio between the case in which the developer was in good operation condition and the case in which the developer was in an insolvent condition. The alternative hypothesis indicated no difference. As shown in Appendix C, Figure A6, since Levene's sig value was 0.359, which was more than 0.05, it can be considered that dispersion homogeneity was assumed. In this case, since the sig value of the *t*-test was 0.976, it was less than 0.05, so the null hypothesis was rejected. Therefore, there was no significant difference between the case in which the developer was in good condition and the case in which the developer was in an insolvent condition.

4.3.2. Builder's Insolvency

This study investigated whether there was a difference in the ratio of defect repair costs depending on the operation ability of the builders of the home construction project. There were very few cases in which the builder was insolvent. The defect repair cost ratio in the operational state was slightly lower than that in the case of insolvency. On the other hand, it was analyzed that there was no statistically significant difference in the results of the *t*-test for the average defect repair cost ratio.

As shown in Figure 11, the builder's number in normal conditions was 286 cases, and the defect repair cost ratio was 0.537%, whereas four cases were for insolvent builders with a defect repair ratio of 0.580%. In the *t*-test analysis, the null hypothesis indicated a difference in the defect repair cost ratio between the case in which the builder was in an operation state and the case in which the builder was in an insolvent state, and the alternative hypothesis showed no difference. As shown in Figure A7, since Levene's sig value was 0.252, which was more significant than 0.05, it could be considered that dispersion homogeneity was assumed. In this case, since the sig value of the *t*-test was 0.799, which was less than 0.05, the null hypothesis was rejected. Therefore, it was judged that there was no significant

difference between the case where the builder was in a normal state and the case where it was in an insolvent state.

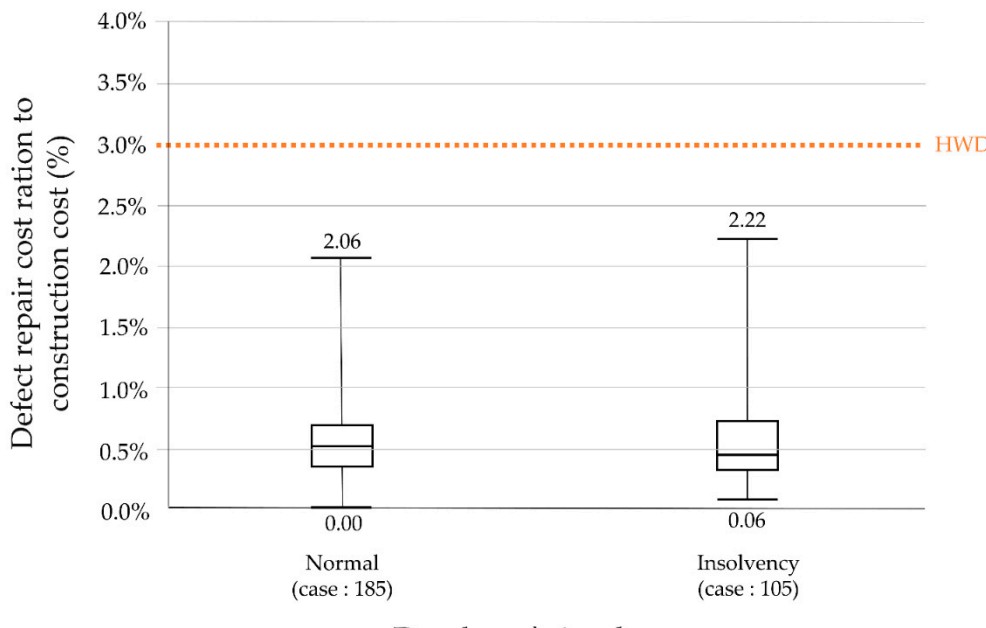

**Figure 10.** Comparison of developer's insolvency.

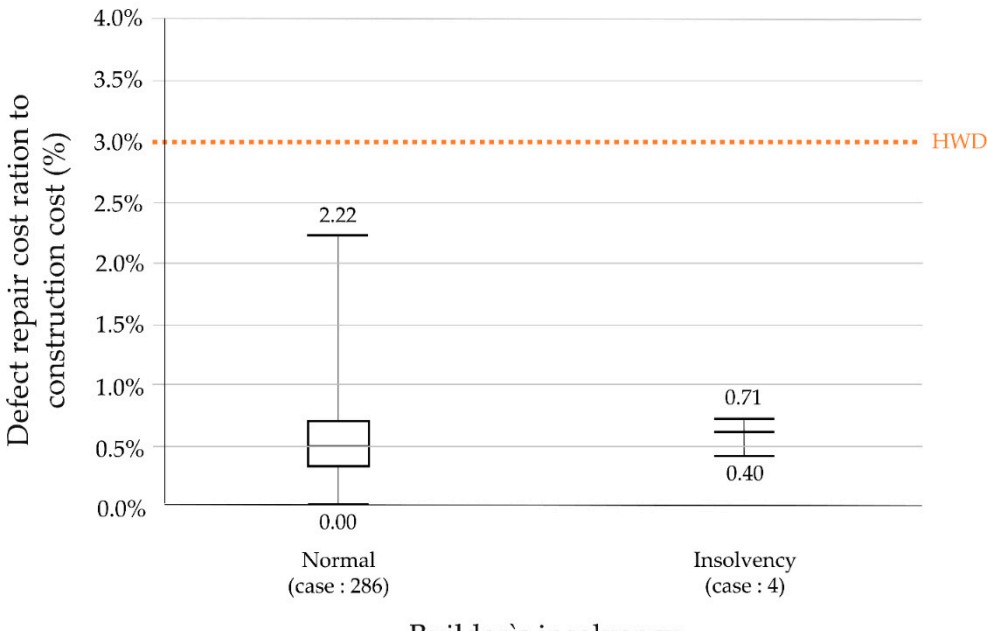

**Figure 11.** Comparison of builder's insolvency.

### 4.3.3. Shared Condition to Defect Repair Cost

This study investigated whether there is a difference in the ratio of repair costs according to the method of sharing the repair costs confirmed in the defect litigation among the participating parties in the home construction project. In most of the cases, the parties shared the cost jointly. The defect repair cost ratio was lower when the warranty company paid the entire repair cost compared to the other cases. Additionally, according to the *t*-test on the average defect repair cost ratio, it was analyzed that the defect repair cost ratio was lower when the warranty company paid all of it compared to the case of mutual sharing.

As shown in Figure 12, there were 18 cases where the developer paid all the repair costs, and the defect repair cost ratio was 0.512%. In the case of mutual sharing, there were 247 cases, and the defect repair cost ratio was 0.554%. In the case of 25 cases covered by the warranty company, the defect repair cost ratio was 0.394%. Therefore, the defect repair cost ratio was the lowest when the warranty company covered all the defect repair costs.

In the ANOVA analysis, the null hypothesis indicated having a difference in the defect repair cost ratio for each case, while the alternative hypothesis stated no difference. The result of the ANOVA analysis, as shown in Figure A8, showed that since Levene's sig value was 0.258, which was more significant than 0.05, it can be considered that dispersion homogeneity was assumed. Additionally, since the null hypothesis was rejected because the sig value of ANOVA was 0.066 or greater than 0.05, it was judged that there was no difference in the defect repair cost ratio between the three methods for defect repair cost-sharing.

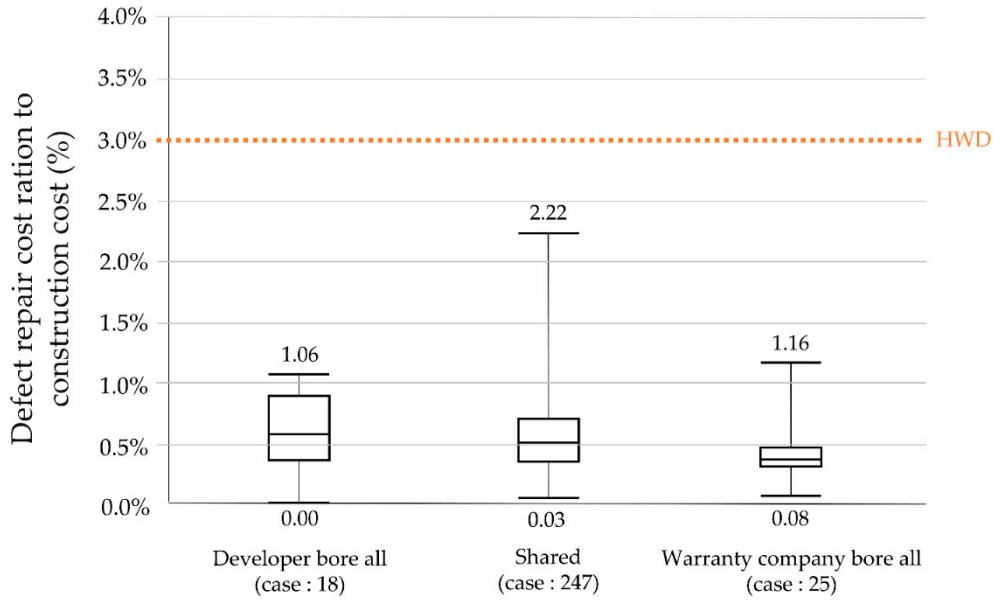

**Figure 12.** Comparison of damage sharing by the project entities.

However, considering that the defect repair cost ratio was relatively low when the warranty company fully bore the cost, it seemed that additional analysis was necessary. Therefore, the *t*-test was used to compare the case of mutual sharing and the case of all sharing by the warranty company. The null hypothesis established that there was a difference in the defect repair cost ratio between the case of mutual sharing and the case of all sharing by the warranty company, whereas the alternative hypothesis established that there was no difference. As can be seen in the results shown in Figure A8, since Levene's sig value was 0.129, which was more significant than 0.05, it could be considered that dispersion homogeneity was assumed. In this case, the sig value of the *t*-test was 0.021, which was smaller than 0.05, so the null hypothesis was accepted. Therefore, the repair cost ratio was 0.554% in the mutual sharing and 0.394% in the case covered by the warranty company, so it could be interpreted that the defect repair cost ratio was lower when the warranty company paid all of it.

### 4.3.4. Additional Damage Sharing by The Warranty Company

In the case of defect litigation, it was examined whether there was a difference in the ratio of the defect repair cost depending on whether there was an additional cost to the warranty company alone. There were more cases where the warranty company had to pay extra. Nevertheless, in this case, the defect repair cost ratio was lower. On the other hand,

it was found that there was a statistically significant difference in the results of the *t*-test on the average defect repair cost ratio.

As shown in Figure 13, there were 170 cases in which the warranty company paid additionally, and the defect repair cost ratio was 0.499%, whereas there were 120 cases where there was no additional burden, and the defect repair cost ratio was 0.593%. Therefore, the ratio of defect repair costs was lower when there was an additional burden. In the *t*-test analysis of these, the null hypothesis established that there was a difference in the defect repair cost ratio between the case where the warranty company bore additional burden alone and the case where there was no additional burden, and the alternative hypothesis set no difference. As shown in Figure A9, since Levene's sig value was 0.303, which was more significant than 0.05, it can be considered that dispersion homogeneity was assumed. In this case, since the sig value of the *t*-test was 0.016, which was less than 0.05, the null hypothesis was adopted. Therefore, it could be interpreted that there was a statistically significant difference between the case where the warranty company made the additional contribution and the case where there was no additional share. Therefore, the defect repair cost ratio was 0.499% for cases with additional burden and 0.593% for cases without additional burden. The defect repair cost ratio might be lower when the warranty company bears additional costs.

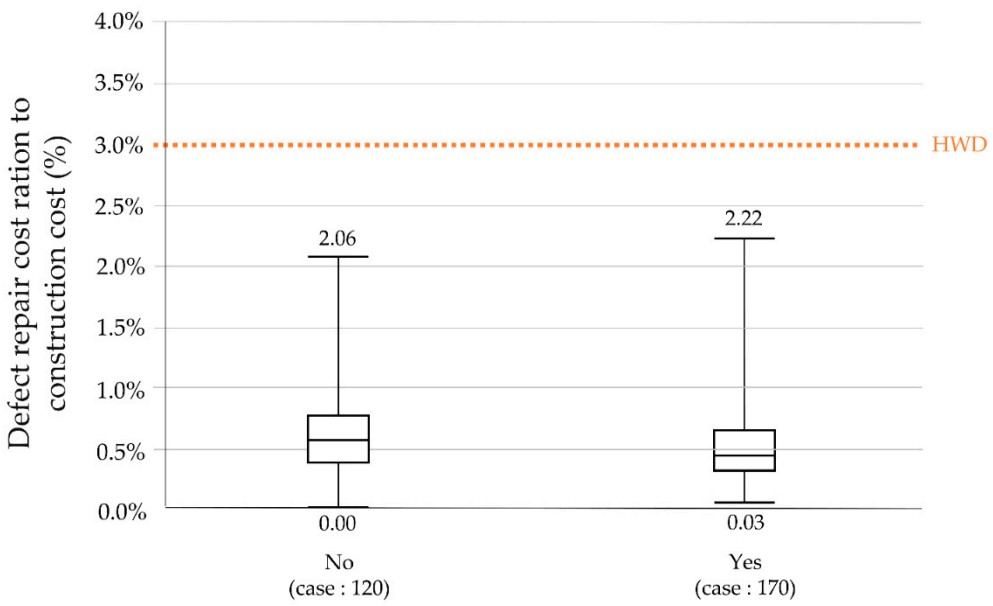

**Figure 13.** Comparison of additional damage sharing by the warranty company.

## 5. Discussion

### 5.1. Reasonable Standard of Home Warranty Deposit

This study investigated the adequacy of the criteria for calculating the home warranty deposit. Since the home warranty deposit is given in preparation for defect repair, knowing the defect repair cost will help calculate the appropriate standard for a home warranty deposit. In addition, considering that the Korean Housing Act sets the defect repair deposit to 3% of the construction cost and many preceding studies listed in Table 1 use the ratio to the construction cost as a measure to evaluate the quality of the home or the level of the repair cost, the ratio of defect repair cost to construction cost was set as a criterion for judgment.

This study was conducted on houses only based on repair costs recognized through litigation for defects that occurred after the house's handover. According to the results, the defect repair cost ratio of the total 290 home complexes was 0.538% on average, 0% at the minimum, and 2.22% at the maximum. This was lower than the results of all other

preceding studies, except for Love et al. [32]. The level was lower than that of Choi [30] and Liu et al. [33], who particularly studied only homes, as in this study. Moreover, in all cases, the repair cost was less than 3% of the construction cost, which is the standard of the Housing Act, so 3% of the construction cost, which is the current standard for defect repair deposits, seems excessive. No law or system can protect everything against all cases. Therefore, it is difficult to provide 100% protection against defects in the home. Considering the above fact, 93.1% of all cases are protected if the defect repair deposit standard is adjusted to about 1% of the construction cost considering the average of 0.538% of the analysis result. Furthermore, even if it is set to 1.5%, as in Choi's opinion [30] among the preceding studies, 98.28% of all cases can be protected, so this is also not unreasonable as a legal standard. Even if the defect repair deposit standard exceeds 1% or 1.5% of the construction cost, it does not mean that all the houses are not protected. For example, suppose that the standard of the warranty deposit is adjusted to 1.5% of the construction cost, as in Figure 2, which shows the defect repair cost ratios in the five cases where the defect repair cost ratio exceeded 1.5% were 1.57%, 1.86%, 1.89%, 2.06%, and 2.22%, respectively. Therefore, as much as 0.07%, 0.36%, 0.39%, 0.56%, and 0.72% of the portion whose warranty deposits exceed 1.5%, respectively, are not protected by the deposit. Instead, it is possible to develop an alternative to protect these excess portions by providing a separate warranty insurance system or by placing conditions under a special contract in the existing warranty insurance. In this way, it can be a measure to strengthen the consumer protection function while preventing the social waste caused by setting excessive deposits.

On the other hand, rather than a method to collectively adjust the standard for warranty deposits as presented above, dividing and adjusting the standard of the home warranty deposit by the scale construction cost to an appropriate level may also be considered. For this purpose, the case of a home construction cost of less than 800 billion KRW was classified into Group A, the case of 800 billion KRW to 1.6 trillion KRW into Group B, and the case of 1.6 trillion KRW or more into Group C. The results showed that the defect repair cost ratio of Group A with a small construction cost was high. On the contrary, Group C's defect repair cost ratio with a considerable construction cost was low. There was also a difference in the mean value, and it was confirmed that it was statistically significant in the ANOVA result. Additionally, there was a difference in each group's defect repair cost ratio distribution. In Group A, there were cases where the defect repair cost ratio exceeded 2%, but all cases belonging to Group B were less than 1.5%, while all cases belonging to Group C were below 1%. Therefore, the defect repair cost ratio distribution varies according to the construction cost scale. As discussed above, the warranty deposit ratio in Australia is differentially applied according to the size of the down payment [40]. In Japan, this category is not based on the construction cost, but the system makes deposits differ according to the number of households [43]. Considering the above facts, it can be a reasonable alternative to vary the housing defect repair deposit according to the scale of the construction cost. Therefore, when considering the scale of the construction cost, it might be reasonable if the warranty deposit standard is set to 2.5% of the construction cost when the construction cost is less than 800 billion KRW, 1.5% of the construction cost when the construction cost is between 800 billion KRW and 1.6 trillion KRW, and 1% of the construction cost when the construction cost is 1.6 trillion KRW or more. Particularly in this case, since it is possible to protect all cases with an adjusted standard, it is advantageous in consumer protection rather than a lump sum adjustment of 1% or 1.5% as suggested above.

*5.2. Influence on Project Conditions*

This study also examined whether the implementation conditions of the home construction project affected the quality or defect repair cost. Various scales have been suggested in prior studies, and in this study, the four following cases were reviewed.

First, it was compared to whether there was a difference according to the public, private, or typical type of project execution entity. According to the analysis result, the defect repair cost ratio of the private sector was 0.525%, and that of the public sector

was 0.672%, which was lower in the private sector, implying that this approach was also statistically significantly different. This means that the private sector has relatively fewer defects than the public sector, which is thought to have impacted quality. All the homes subject to the case analysis of this study are presale homes. In addition to rental homes, homes for sale are built and supplied in large quantities in Korea. This is different from the supply of rental homes by the public sector in other countries. Depending on the object, the construction project differs in various aspects when done by the public or the private sector. However, in the home construction industry, in many countries, the most important policy goal for the public sector is to supply homes continuously and stably. On the other hand, the private sector has to be chosen by consumers to generate sustainable profits, so it has to consider satisfying quality. Therefore, the case analysis results can be understood in this context. This analysis result is different from Martin and Westerhoff [54], where there was no difference in quality between the undertaking entity of public and private sectors, and it was the exact opposite result from the study by Forcada et al. [31] that claimed that the public supply method was superior to the private because the rework cost was less. Therefore, it seems that the public sector in the home construction project should make an effort to reduce defects compared to the private sector.

Second, it was investigated whether there is a difference between the trust implementation, the direct implementation of the landlord, and the joint implementation according to the land and financial procurement method for project implementation. According to the analysis results, the defect repair cost ratio by the trust implementation was 0.528%, landlord implementation was 0.535%, and joint implementation was 0.584%. If the defect repair cost ratio is compared, the trust implementation has the lowest defect repair cost ratio, which may be advantageous in terms of quality. However, the defect repair cost ratio difference between the trust implementation and the landlord implementation was insignificant, and the ANOVA analysis result showed no statistically significant difference. Therefore, it might be difficult to conclude that the quality of the trust implementation method in the home construction project is superior to that of other methods based on the analysis results alone. Trust companies have differentiated experience and capabilities in reviewing and managing business feasibility, including financial procurement. On the other hand, landlords generally lack expertise compared to trust companies. However, the analysis subject included cases where large construction companies and specialized developers were landlords. In this case, they are superior to general landlords and have financial power and business management ability comparable to a trusted company. Considering these points, it is considered that the landlord implementation project cannot be significantly lower in quality than the trust project. Therefore, it implies that quality difference can be unreliable; this is in contrast to the views of Cotter and Richard [55], Topuz and Isik [56], and Grybauskas and Pilinkiene [57], who emphasized the superiority of trust implementation, or Ambrose et al. [58], Vogel [59], and Kawaguchi et al., who emphasized the limitations of trust implementation.

Third, the comparison was reviewed between the case in which the developer directly constructed the homes and the case in which a contract was made. According to the analysis results, the defect repair cost ratio for the owner project was 0.585% and for the contract project 0.523%, so it seems that the quality of the contract project is relatively good. According to the *t*-test result, since it was challenging to say that there was a statistically significant difference, one cannot conclude a significant quality difference between the two. In general, it is expected that the quality of the construction will be excellent if a construction company with abundant experience and expertise in the construction field directly performs construction. According to the analysis result, the defect repair cost ratio of the contract project was slightly lower, which could be in line with the views of Neap and Aysal [63], Serpell [64], and Hwang et al. [29]. In particular, according to a study by Hwang et al., the number of cases of developers and contractors was 5.4% and 2.2%, respectively, so there was a difference of more than two times. Moreover, it is unclear whether homes were included, but in the case of building construction, the difference was wider at 4.6% and

0%, respectively. Of course, since this was a case in which the developer and the builder each reworked, it was difficult to regard it as the perfect case of repairing defects in this study. However, according to the results of this study, it does not appear that there is such a big difference, so the argument that the direct management method as in Zou [61] and Korkmaz et al. [62] could secure sufficient quality through the interest and investment of the developer was still considered as valid.

Fourth, the difference between the case of a single prime contract for construction and the case of multiple prime contracts was examined. According to the analysis results, the defect repair cost ratio for a single prime contract was 0.508%, whereas it was 0.596% for multiple prime contracts, so there seemed to be a difference in values. Additionally, it was analyzed that there was a statistically significant difference in the *t*-test result. Therefore, it can be interpreted that a single prime contract is advantageous in terms of quality because the ratio of defect repair cost was lower in a single prime contract than in multiple prime contracts, for housing construction work. This is consistent with the views of Nooteboom [65] and Al-Hammad [66], who emphasized that a single prime contract was more efficient than multiple prime contracts. In addition, the results of this study can be used as the basis for the fact that a single prime contract can be more advantageous than multiple prime contracts in terms of quality in the home construction project.

*5.3. Influence on Lawsuit Issues*

A home warranty deposit for home construction is used to prepare for the case where the project owner becomes insolvent and cannot properly perform defect repairs. Therefore, it was considered whether the project owner was insolvent, and if so, how much repair cost would be required. If it is found that there is a high risk that the project owner will be insolvent and the cost of repairing defects is found to be exceptionally high, countermeasures should be devised accordingly.

However, according to the analysis results, 105 out of 290 cases of project owners were insolvent, accounting for 36.21% of the total. On the contrary, the number of cases where the builders became insolvent was four, accounting for 1.38% of the total. Therefore, it was found that the possibility that the developer would become insolvent was very high compared to the builders. The developer is directly responsible for the homeowner's claim for repair. However, defect repair would not be executed properly if the developer was insolvent. If this happens, the possibility of a dispute increases, and in Korea, it escalates to defect litigation. Not only does litigation waste a substantial amount of time and money, but it also damages the reputation of the home construction industry, which is a considerable blow to homeowners and builders regardless of the lawsuit's outcome. As mentioned above, because the risk of bankruptcy in the construction and real estate industries is being reported worldwide [20–26], and the insolvency of the developers remains in the analysis results, it is considered that social institutions to protect homeowners such as home warranty deposits are necessary.

One may examine whether there was a difference in the defect repair cost ratio depending on whether the project entities were in good operating conditions. The percentage of defect repair costs according to the capability of the project owner was 0.538% and 0.539% in the insolvent state, which showed little difference between the two. The *t*-test results also showed that there was no statistically significant difference. Therefore, there was no difference in the defect repair cost ratio depending on whether the project owner was capable. According to the project owner's capability, the defect repair cost ratio was 0.537% in the operational state and 0.580% in the insolvent state. Therefore, the defect repair cost ratio was slightly higher when the project owner was insolvent. However, the *t*-test results showed no statistically significant difference between them, so there was no difference in the defect repair cost ratio according to the self-efficacy of the project owner. In summary, there is no need to add a special home warranty deposit even if the developer or builder, who is the project owner, becomes insolvent.

Meanwhile, one may examine whether there was an additional burden to the warranty company while having the project entities share the cost of repairing defects in a lawsuit. In litigation, the liability of business entities to bear the cost of repairing defects is reduced for various reasons. For example, if specific data confirm that the project entities have repaired defects, these data include the time, location, portion of the defect, and extent of the damage. In addition, the time when the repair is performed and the materials and methods used for the repair are recorded, while photos and videos that can be compared before and after the repair should be included. In addition to this, data such as a confirmation letter from homeowners acknowledging that the project entity has completed repairing defects must be presented. In this case, the court may, ex officio, reduce some of the responsibilities of the project entity. In addition, if it is evident that the warranty period for defects has expired when considering the period of use after handover, the repair cost for the defect may be reduced by the period [70]. However, apart from the part where the responsibility of the project entity is reduced in this way, some of the parts that have already been reduced in consideration of the defect compensation warranty system's characteristics are considered a warranty liability, and the warranty company bears additional burdens. This is a logical system from the point of view of jurisprudence, and it is beyond the scope of this study to identify it as it cannot be quantified because it is due to the specificity of each case.

Nonetheless, this measure protects the rights of homeowners. Therefore, in this study, parties do not make a legal judgment on the cost of repairing defects to be additionally borne by the warranty company. However, one may only look at how much this defect repair cost was in terms of cost and whether the difference was excessive. The previous analysis results revealed an additional burden on the warranty company in 170 cases or 58.6% of the total. However, the defect repair cost ratio was 0.593% when there was no additional burden on the warranty company, whereas the defect repair cost ratio was lower at 0.499% when the warranty company was paid additionally. In other words, the defect repair cost ratio was lower even though the court considered the warranty liability to be heavier and placed an additional burden on the warranty company. It can also be seen that there were fewer defects in such cases. In addition, the institutional device for the protection of homeowners plays a valuable role.

## 6. Conclusions

Home defects are becoming a social problem all over the world. In most countries, the developer or builder, the primary entity of the home construction project, has to repair the defect. However, cases in which project owners are in insolvency where they cannot perform defect repairs, such as bankruptcy or court management, frequently occur. A home warranty deposit is a support system for such cases. One can reasonably prepare a warranty deposit if one knows how much it will cost to repair a home defect.

In this study, how much defect repair costs were compared to construction costs was investigated using data on repair costs concluded in the lawsuits against 290 home complexes in Korea. If the actual defect repair cost was more or less than the home warranty deposit, accordingly, it was necessary to adjust the warranty deposit to an appropriate level. In addition, the differences according to the mode of project execution and litigation issues were compared. The difference in the defect repair cost according to the different review subjects would help prepare counteraction such as quality control to reduce defects.

The analysis results showed that the cost of repairing defects in the Korean home construction project was 0.538% of the construction cost, which was lower than the 3% of the construction cost stipulated by the Korean Housing Act. In 93.1% of all cases, the defect repair cost to construction cost was below 1%. Therefore, even if the home warranty deposit is adjusted to about 1% of the construction cost, it will likely be pleasing. The comparison result showed that for the defect repair cost ratio, according to the construction cost and from the viewpoint of supplementing this, cases with a construction cost of less than 800 billion KRW should require less than 2.5% of the construction cost; cases with a construction cost of 800 billion KRW to 1.6 trillion KRW should require less than 1.5% of the

construction cost; and a construction cost of 1.6 trillion KRW or more should require below 1% of the construction cost. Therefore, if the defect repair deposit is applied differentially according to the scale of the construction cost, it would be more effective than the previous standard because it can protect against all cases.

While pursuing the precedent studies about the defect repair cost of the home, it was found that either the number of cases was limited or the cases that were not homes were included. In this study, all the cases were related to the homes, and the cases were collected in a sufficient quantity to propose the analysis results. Therefore, it is expected that the study results could be utilized to adjust the home warranty deposit system and rationalize the housing industry's policy. Though it was not expected to have an entirely different defect type in the single home or dwelling housing, the results should be based on the dwelling homes since this study targeted them. In addition, it is essential to research the defect cause analysis and the types and factors affecting the defect. Accordingly, it is necessary to discuss the difference between the defect repair cost and the plan to prevent defects.

Meanwhile, in the home construction project, since the private sector had a lower rate of repair costs than the public sector, an overall effort to improve the quality of the public sector is necessary. In addition, institutional supplementation for quality improvement is required in the case of multiple prime contracts since it was analyzed that a single prime contract had a lower defect repair cost than multiple prime contracts. There was also a higher risk if the developers became insolvent instead of the builders as far as the project entities were concerned. There was no difference in quality due to the incompetence of the project implementor; however, the incompetence of the project owner can become a social issue when it is necessary to prepare for a downturn caused by the business cycle. The joint efforts and cooperation of researchers urgently need to be examined in follow-up studies regarding these agendas.

Lastly, as this study focused on the standard of home warranty deposit, it had the following limitations. First, there might be other criteria for a home warranty deposit other than construction cost. In Korea, the home warranty deposit is calculated based on the construction cost, and the prior studies also compared the defect repair cost with the construction cost. Therefore, comparing the housing defect repair deposit and construction cost did not seem unreasonable. This study was limited to comparing the construction costs as it judged the adequacy of the home warranty deposit standard using the Korean case. However, some countries, such as Canada and Japan, operate a defect repair deposit system based on the number of households. In addition, the area is used as an essential criterion for buildings such as homes. As discussed above, various criteria serve as crucial standards for construction, and it is presumed that the defect repair cost and the home warranty deposit are not irrelevant. Therefore, it is necessary to compare the defect repair cost and the home warranty deposit according to criteria such as the number of households and area in addition to the construction cost. Second, since the construction cost, the number of households, and the area of each home are different, it is estimated that the construction cost per area or construction cost per household will also differ. In other words, it is considered that there is a difference in the planned quality because the level of construction cost input for each home is different. It can also be a good research topic to compare the relationship between the planned quality and the defect, called the post quality.

**Author Contributions:** Conceptualization, J.P. and D.S.; methodology, J.P. and D.S.; software, J.P.; validation, J.P. and D.S.; formal analysis, J.P.; investigation, J.P.; resources, J.P.; data curation, J.P.; writing—original draft preparation, J.P.; writing—review and editing, J.P. and D.S.; visualization, J.P.; supervision, D.S.; project administration, D.S; funding acquisition, D.S.; All authors have read and agreed to the published version of the manuscript.

**Funding:** This research was funded by the National Research Foundation of Korea 2019R1A2C1009913. And The APC was funded by the National Research Foundation of Korea.

**Institutional Review Board Statement:** Not applicable.

**Informed Consent Statement:** Not applicable.

**Data Availability Statement:** The data presented in this study are available on request from the corresponding author.

**Conflicts of Interest:** The authors declare no conflict of interest.

## Appendix A

### Descriptives

| | N | Mean | Std. Deviation | Std. Error | 95% Confidence Interval for Mean | | Minimum | Maximum |
|---|---|---|---|---|---|---|---|---|
| | | | | | Lower Bound | Upper Bound | | |
| Less than 80 billion KRW | 117 | 0.00067308 | 0.000400978 | 0.000037070 | 0.00059965 | 0.00074650 | 0.000094 | 0.002218 |
| 80~160 billion KRW | 92 | 0.00050038 | 0.000242360 | 0.000025268 | 0.00045019 | 0.00055057 | 0.000000 | 0.001467 |
| More than 160 billion KRW | 81 | 0.00038591 | 0.000214249 | 0.000023805 | 0.00033854 | 0.00043329 | 0.000030 | 0.000931 |
| Total | 290 | 0.00053808 | 0.000331750 | 0.000019481 | 0.00049974 | 0.00057643 | 0.000000 | 0.002218 |

### Test of Homogeneity of Variances

| Levene Statics | df1 | df2 | Sig. |
|---|---|---|---|
| 11.438 | 2 | 287 | 0.000 |

### ANOVA

| | Sum of squares | df | Mean square | F | Sig. |
|---|---|---|---|---|---|
| Between groups | 0.000 | 2 | 0.000 | 21.464 | 0.000 |
| Within groups | 0.000 | 287 | 0.000 | | |
| Total | 0.000 | 289 | | | |

**Figure A1.** ANOVA result of each group to construction cost scale.

## Appendix B

### Group Statistics

| | | N | Mean | Std. Deviation | Std. Error Mean |
|---|---|---|---|---|---|
| Defect repair cost ratio to construction cost | Private sector | 264 | 0.005251 | 0.0033102 | 0.0002037 |
| | Public sector | 23 | 0.006722 | 0.0033363 | 0.0006957 |

### Independent Samples Test

| | | Levene`s Test for Equality of Variances | | t-test for Equality of Means | | | | | | | |
|---|---|---|---|---|---|---|---|---|---|---|---|
| | | F | Sig. | t | df | Sig. (2-tailed) | Mean Difference | Std. Error Difference | 95% Confidence interval of the difference | |
| | | | | | | | | | Lower | Upper |
| Defect repair cost ratio to construction cost | Equal variances assumed | 0.226 | 0.635 | -2.042 | 285 | 0.042 | -0.0014706 | 0.0007201 | -0.0028880 | -0.0000532 |
| | Equal variances not assumed | | | -2.029 | 25.919 | 0.053 | -0.0014706 | 0.0007249 | -0.0029608 | 0.0000196 |

**Figure A2.** *t*-test result of project entities.

## Descriptives

| | N | Mean | Std. Deviation | Std. Error | 95% Confidence Interval for Mean | | Minimum | Maximum |
|---|---|---|---|---|---|---|---|---|
| | | | | | Lower Bound | Upper Bound | | |
| Trust project | 20 | 0.005280 | 0.0029214 | 0.0006532 | 0.003913 | 0.006647 | 0.0008 | 0.0117 |
| Landlord project | 244 | 0.005348 | 0.0033213 | 0.0002126 | 0.004930 | 0.005767 | 0.0000 | 0.0222 |
| Joint project | 26 | 0.005742 | 0.0036729 | 0.0007203 | 0.004259 | 0.007226 | 0.0003 | 0.0186 |
| Total | 290 | 0.005379 | 0.0033191 | 0.0001949 | 0.004995 | 0.005763 | 0.0000 | 0.0222 |

## Test of Homogeneity of Variances

| Levene Statics | df1 | df2 | Sig. |
|---|---|---|---|
| 0.045 | 2 | 287 | 0.956 |

## ANOVA

| | Sum of squares | df | Mean square | F | Sig. |
|---|---|---|---|---|---|
| Between groups | 0.000 | 2 | 0.000 | 0.174 | 0.840 |
| Within groups | 0.003 | 287 | 0.000 | | |
| Total | 0.003 | 289 | | | |

**Figure A3.** ANOVA result of project implementation type.

## Group Statistics

| | | N | Mean | Std. Deviation | Std. Error Mean |
|---|---|---|---|---|---|
| Defect repair cost ratio to construction cost | Contract Project | 221 | 0.005233 | 0.0030710 | 0.0002066 |
| | Own Project | 69 | 0.005845 | 0.0040025 | 0.0004818 |

## Independent Samples Test

| | | Levene's Test for Equality of Variances | | t-test for Equality of Means | | | | | | | |
|---|---|---|---|---|---|---|---|---|---|---|---|
| | | F | Sig. | t | df | Sig. (2-tailed) | Mean Difference | Std. Error Difference | 95% Confidence interval of the difference | |
| | | | | | | | | | Lower | Upper |
| Defect repair cost ratio to construction cost | Equal variances assumed | 3.840 | 0.052 | -1.338 | 288 | 0.182 | -0.0006114 | 0.0004571 | -0.0015111 | 0.0002882 |
| | Equal variances not assumed | | | -1.166 | 94.310 | 0.246 | -0.0006114 | 0.0005243 | -0.0016523 | 0.0004294 |

**Figure A4.** *t*-test result of building execution.

**Group Statistics**

| | | N | Mean | Std. Deviation | Std. Error Mean |
|---|---|---|---|---|---|
| Defect repair cost ratio to construction cost | Single Prime Contract | 192 | 0.005081 | 0.0029876 | 0.0002156 |
| | Multiple Prime Contract | 98 | 0.005963 | 0.0038375 | 0.0003876 |

**Independent Samples Test**

| | | Levene`s Test for Equality of Variances | | t-test for Equality of Means | | | | | | | |
|---|---|---|---|---|---|---|---|---|---|---|---|
| | | F | Sig. | t | df | Sig. (2-tailed) | Mean Difference | Std. Error Difference | 95% Confidence interval of the difference | |
| | | | | | | | | | Lower | Upper |
| Defect repair cost ratio to construction cost | Equal variances assumed | 5.660 | 0.018 | -2.155 | 288 | 0.032 | -0.0008825 | 0.0004095 | -0.0016885 | -0.0000766 |
| | Equal variances not assumed | | | -1.990 | 158.593 | 0.048 | -0.0008825 | 0.0004436 | -0.0017586 | -0.0000065 |

**Figure A5.** *t*-test result of building contract.

## Appendix C

**Group Statistics**

| | | N | Mean | Std. Deviation | Std. Error Mean |
|---|---|---|---|---|---|
| Defect repair cost ratio to construction cost | Normal | 185 | 0.005375 | 0.0032157 | 0.0002364 |
| | Insolvency | 105 | 0.005387 | 0.0035097 | 0.0003425 |

**Independent Samples Test**

| | | Levene`s Test for Equality of Variances | | t-test for Equality of Means | | | | | | | |
|---|---|---|---|---|---|---|---|---|---|---|---|
| | | F | Sig. | t | df | Sig. (2-tailed) | Mean Difference | Std. Error Difference | 95% Confidence interval of the difference | |
| | | | | | | | | | Lower | Upper |
| Defect repair cost ratio to construction cost | Equal variances assumed | 0.845 | 0.359 | 0.030 | 288 | 0.976 | 0.0000121 | 0.0004063 | -0.0007875 | 0.0008117 |
| | Equal variances not assumed | | | 0.029 | 200.931 | 0.977 | 0.0000121 | 0.0004162 | -0.0008086 | 0.0008327 |

**Figure A6.** *t*-test result of developer's insolvency.

### Group Statistics

| | | N | Mean | Std. Deviation | Std. Error Mean |
|---|---|---|---|---|---|
| Defect repair cost ratio to construction cost | Normal | 286 | 0.005373 | 0.0033389 | 0.0001974 |
| | Insolvency | 4 | 0.005800 | 0.0014024 | 0.0007012 |

### Independent Samples Test

| | | Levene`s Test for Equality of Variances | | t-test for Equality of Means | | | | | | | |
|---|---|---|---|---|---|---|---|---|---|---|---|
| | | F | Sig. | t | df | Sig. (2-tailed) | Mean Difference | Std. Error Difference | 95% Confidence interval of the difference | |
| | | | | | | | | | Lower | Upper |
| Defect repair cost ratio to construction cost | Equal variances assumed | 1.319 | 0.252 | 0.255 | 288 | 0.799 | 0.0004269 | 0.0016738 | -0.0028676 | 0.0037215 |
| | Equal variances not assumed | | | 0.587 | 3.494 | 0.594 | 0.0004269 | 0.0007285 | -0.0017164 | 0.0025703 |

**Figure A7.** *t*-test result of builder's insolvency.

### Descriptives

| | N | Mean | Std. Deviation | Std. Error | 95% Confidence Interval for Mean | | Minimum | Maximum |
|---|---|---|---|---|---|---|---|---|
| | | | | | Lower Bound | Upper Bound | | |
| Developer bore all the costs | 18 | 0.005122 | 0.0033158 | 0.0007816 | 0.003473 | 0.006771 | 0.0000 | 0.0106 |
| Shared | 247 | 0.005543 | 0.0033641 | 0.0002141 | 0.005122 | 0.005965 | 0.0003 | 0.0222 |
| Warranty company bore all the costs | 25 | 0.003940 | 0.0025267 | 0.0005053 | 0.002897 | 0.004983 | 0.0008 | 0.0116 |
| Total | 290 | 0.005379 | 0.0033191 | 0.0001949 | 0.004995 | 0.005763 | 0.0000 | 0.0222 |

### Test of Homogeneity of Variances

| Levene Statics | df1 | df2 | Sig. |
|---|---|---|---|
| 1.361 | 2 | 287 | 0.258 |

### ANOVA

| | Sum of squares | df | Mean square | F | Sig. |
|---|---|---|---|---|---|
| Between groups | 0.000 | 2 | 0.000 | 2.739 | 0.066 |
| Within groups | 0.003 | 287 | 0.000 | | |
| Total | 0.003 | 289 | | | |

**Figure A8.** ANOVA result of damage sharing to project entities.

**Group Statistics**

| | | N | Mean | Std. Deviation | Std. Error Mean |
|---|---|---|---|---|---|
| Defect repair cost ratio to construction cost | Shared | 247 | 0.005543 | 0.0033641 | 0.0002141 |
| | All covered by surety company | 25 | 0.003940 | 0.0025267 | 0.0005053 |

**Independent Samples Test**

| | | Levene`s Test for Equality of Variances | | t-test for Equality of Means | | | | | | | |
|---|---|---|---|---|---|---|---|---|---|---|---|
| | | F | Sig. | t | df | Sig. (2-tailed) | Mean Difference | Std. Error Difference | 95% Confidence interval of the difference | |
| | | | | | | | | | Lower | Upper |
| Defect repair cost ratio to construction cost | Equal variances assumed | 2.313 | 0.129 | 2.316 | 270 | 0.021 | 0.0016033 | 0.0006922 | 0.0002404 | 0.0029662 |
| | Equal variances not assumed | | | 2.921 | 33.281 | 0.006 | 0.0016033 | 0.0005488 | 0.0004871 | 0.0027195 |

**Figure A9.** *t*-test result of additional damage sharing to the warranty company.

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
