# Peer review of "Defect Repair Cost and Home Warranty Deposit, Korea"

_buildings, doi:10.3390/buildings12071027_

Round 1

Reviewer 1 Report

Dear Authors, 

Good day to you.  The manuscript is well written and discussed Defect Repair Cost and Home Warranty Deposit, Korea, Best wishes to all the authors. The methodology part can be modified for a better understanding. The conclusion can be rewritten for enhancing the paper. 

Best Regards

Author Response

Dear reviewer

Thank you very much for reviewing our paper despite your busy schedule.

The conclusion was adjusted according to the recommendation by the reviewer.

  • The overall text was amended in Chapter 3, which covered the methodology.
  • The study achievement was added in Conclusion, the limitation in the present study was disclosed, and the future research prospect was discussed accordingly.

Reviewer 2 Report

Review Report

This manuscript deal with the litigation details of 290 home complexes in Korea to investigate defect repair costs and factors affecting them. According to the analysis results, the defect repair cost was 0.538% of the construction cost on average. The authors' claim proposes a method to adjust the warranty deposit collectively, and a means to apply it differentially according to the size of the construction cost. This is a fascinating manuscript, and I think Buildings readers will be interested.

The presentation is adequate; I have detected some criticisms in the text that should be properly addressed.

The Authors can benefit from the comments below to improve their paper.

1.   First, the paper remains rather vague with regard to key terms and concepts, and this is also mirrored in a phrasing that could often be much more precise.

2.   In the introduction section, more new related works of literature are suggested to be referenced.

3.   The main constraints of the research should be discussed in the conclusion section.

4.   The resolution of the figures in the manuscript is not enough, and the authors are advised to improve it.

5.   Research limitations should be mentioned.

6.   The proposed research and obtained results should be compared with some research that was conducted before. In this way, the research implications could be proposed.

7. It is suggested to clarify the construction cost unit of measure shown in Figure 3.

8.   It is recommended to summarize research contributions and reflections in the Conclusion.

9.   Future research direction should be indicated in the Conclusion.

Other suggestions

(1) Please review the entire document for typographical errors and other necessary corrections; check headings, tables, and figures.

(2) Authors should check this manuscript for grammar.

(3) Authors should follow the journal format as follow:

https://www.mdpi.com/journal/buildings/instructions

See the Reference List and Citations Guide for more detailed information.

https://www.mdpi.com/authors/references

In the methodology section, the authors must discuss the methodology carefully.

Author Response

Dear reviewer

Thank you very much for reviewing our paper despite your busy schedule.

We have revised the paper by reflecting the contents below according to the reviewer’s opinion. 
[1] Amended items  
 â‘  The contents of Chapter 3’s case study method were amended, and the terminologies and concepts were clearly defined.  
 â‘¡ The cited literatures in the Introduction were replaced with the latest ones.
 â‘¢â‘¤â‘§â‘¨ The limitation of the present study was disclosed, and a future task was proposed.  
 â‘£The resolution of all the inserted pictures was upgraded to 768dpi.
 â‘¥ The comparison detail with the initial research was described in Results, and the results of the initial studies were mentioned in Discussion.

 Addition 1: The main text was modified according to the format and citation method of the works of literature of the mdpi journal. The attached internet address links were also updated to the latest ones.

 Addition 2: Concerning the methodology, the analysis procedure was added in Chapter 3.3 to explain the analysis and comparison method in Chapter 4.

Reviewer 3 Report

I state that I really appreciate the work done, both for the research carried out and for the results obtained.

I suggest you:

better illustrate how the construction costs of the 290 cases were confirmed

It would be very useful to relate all costs to the unit of measurement (square meter).

This element would make it easier to develop further reflections. For example, only the overall construction cost is considered in Figure 3. The unit cost of construction would allow us to better understand the quality of the buildings (a higher unit cost of construction means better characteristics of the property). Why wasn't it done? To specify.

I have a doubt about the division into groups A, B and C: according to what criteria were the 3 groups formed? Because the subdivision into "Group A for cases less than 800 535 billion KRW, Group B for cases between 800 billion KRW and 1.6 trillion KRW Group C 536 for cases of 1.6 trillion KRW or more". It seems arbitrary. Specify the criteria and reasons.

The subdivision made on the basis of the unitary construction cost of the buildings, would have made it possible to highlight different levels of finish of the buildings and subsequently develop other hypotheses. For example, in buildings with very high unit construction costs, unit repair costs are the same as buildings with low unit construction costs (meaning repairs are similar) but the DRCRR would certainly be lower. If possible, try to make such a reading.

Author Response

Dear reviewer

Thank you very much for reviewing our paper despite your busy schedule.

The main text was amended to reflect the opinion of the reviewer.

â‘  In Korea, 3% of the construction cost is set as the home warranty deposit, and the deposit amount is specified in the judgement ruling of each litigation case. Therefore, the construction cost of the cases in this study was reversely calculated by dividing the warranty deposit by 3%. The detail was added in Chapter 3.2.

â‘¡â‘£ We would be able to assess the quality level if we consider the ‘construction cost per unit area’ suggested by the reviewer. However, this study focused on the calculation standard of the home warranty deposit; therefore, there was a limit that the study was conducted only on the ‘construction cost,’ which is a legal standard. For such limitations, we have added a comment in Conclusion. In addition, we would like to conduct an in-depth analysis in consideration of the ‘construction cost per unit area’ suggested by the reviewer, how much the difference in the ‘construction cost per unit area’ was, and if there was any significant relation with the defect repair ratio for future studies.

â‘¢ Regarding the three types of construction cost category as regards construction cost size, the authors compared the scale, where the difference might have existed by assuming various scales while observing the repair cost distribution against construction cost per case, as the reviewer noted. The construction cost was divided into KRW 500 billion and KRW 1 trillion, but there was no significant difference. Meanwhile, we categorized the construction costs again as below KRW 800 billion (Group A), more than KRW 800 billion to below KRW 1.6 trillion (Group B), and more than KRW 1.6 trillion (Group C). As emphasized in the paper, these three groups showed average differences in ANOVA. Furthermore, the maximum values of defect repair costs against construction costs of these three groups were 2.5%, 1.5%, and 1%, respectively. If the analysis results were considered, there might be a meaning in categorizing these three groups. However, we found that the reason for grouping according to the construction cost was omitted, as the reviewer pointed out. We have added additional detail about it in Chapter 4.1.2. 
